# Dynamic of Tunneled Planing Hulls in Waves

Fatemeh Roshan [1], Sasan Tavakoli [2,3], Simone Mancini [4] and Abbas Dashtimanesh [5,*]

1    Estonian Maritime Academy, Tallinn University of Technology, 11712 Tallinn, Estonia;
     fatemeh.roshan@taltech.ee
2    Department of Mechanical Engineering, Aalto University, 02150 Espoo, Finland; sasan.tavakoli@aalto.fi
3    Department of Infrastructure Engineering, The University of Melbourne, Parkville, VIC 3052, Australia
4    Force Technology, Division for Maritime Industry, Hydro and Aerodynamics Department,
     2800 Kongens Lyngby, Denmark; simo@forcetechnology.com
5    Department of Engineering Mechanics, KTH Royal Institute of Technology, 11428 Stockholm, Sweden
*    Correspondence: abbasda@kth.se

**Abstract:** A tunneled planing craft is a high-speed boat with two tunnels over the hull bottom that are designed to improve the vessel's performance. Hydrodynamic performance of tunneled planing hulls in calm-water is well-known, however, current information on wave conditions is limited. In this study, two different tunneled planing hulls with two degrees of freedom in heave and pitch motions are studied in regular waves by using the computational fluid dynamics (CFD) method based on the Unsteady Reynolds Averaged Navier-Stokes Equations (URANSE) in conjunction with $k-\epsilon$ turbulence model. The results demonstrate that tunneled planing hull motions in waves are nonlinear. In addition, it is found that the dynamic responses of heave and pitch motions as well as occurrence portability of the fly-over phenomenon significantly increases as the Froude number grows. Fly-over motions resulted in vertical motions and acceleration up to 5*g*, high impact pressure, and large induced drag. At a very high planing speed, after flying over the water surface, when the vessel re-enters the water, the resulting hydrodynamic load leads to a second fly-over motion. Since the fly-over is an unwanted movement with adverse effects, these results can provide a better understanding of the fly-over motion that one may consider in future design for improving the planing hull performance.

**Keywords:** tunneled planing hulls; dynamic in waves; fly-over motion; head sea; computational fluid dynamics (CFD)

## 1. Introduction

Tunneled planing hulls are marine vehicles that use aerodynamic pressure to reduce the drag-over-lift ratio of the lifting surface by trapping the airflow in tunnels. The entrapped air increases the dry area of the bottom surface, decreases the wetted area, and gives rise to aerodynamic lift, which partially supports the weight of the vessel. All these factors together enable the boat to operate at higher speeds with less fuel consumption. The use of these vessels has been accelerated in the recent decade and has sparked up some new experimental and numerical research, which has provided a fundamental understanding of their hydrodynamics in calm-water conditions. The use of these vessels is widespread, and there is an urgent need to evaluate their performance in waves, yet very few studies are conducted in this realm.

Generally, planing problems, either steady or unsteady, refers to the advancement of a marine vehicle under the support of hydrodynamic pressure that is generated by the water flowing toward the bottom surface [1–3]. When a high-speed vessel that is operating in planing mode is exposed to water waves, motions in vertical and traverse planes occur [4] and hydrodynamic pressure, which dominantly supports the weight [5], gives rise to nonlinear forces. Consequently, the restoring forces and moments that are generated

by hydrostatic pressure decrease. In addition, the vessel skims on the water surface [6], which is wetting its surface and the area that is washed by water is small and narrow [7]. Moreover, the wetted surface of a planing vessel operating in waves might vary over time. This makes the physics of the problem more complicated. In this condition, the wetted surface is response-dependent, and the added mass along with damping forces/moments is time-dependent.

There is accumulating evidence that wave-induced motions of planing hulls are nonlinear (e.g., [8]). The contribution of hydrodynamic force in support of the weight force is likely to be the reason underlying nonlinearities. The energy corresponding to the motions, including heave and pitch, can be shifted to second and third harmonics, even in the case where the wave steepness is gentle. The nonlinearity of the motions has been seen to occur at wavelengths ranging between $2L$ to $4L$ in head sea conditions (here $L$ refers to the length of the boat). The motion of the vessel resonates over the aforementioned range of wavelengths. The natural frequencies of heave and pitch motions are observed to decrease by increasing the speed of the vessel, while the larger dynamic response is expected to occur by increasing the speed [9]. A wide range of datasets that were measured in towing tanks and the sea has shown that the vertical acceleration of the vessel at its bow is highly increased by the increase of speed [4]. The bow acceleration can reach up to $2g$, where $g$ is the gravitational acceleration, leading to an uncomfortable riding situation for the crew of the vessel. In addition, under the force that is caused by the large vertical acceleration, the structure of the vessel may significantly vibrate, which might get damaged over time.

A vessel operating in the planing mode in rough water can exit the water and re-enter it in a short period, being nearly half of a cycle [10]. Such a phenomenon, identified as fly-over motion, gives an account of large impact forces. The added resistance of the vessels at a higher speed can significantly increase since an induced drag can be influenced by the water waves [11] (i.e., extra hydrodynamic pressure can act on the vessel). The riders of planing hulls frequently prefer to de-accelerate in rough water conditions, preventing extreme responses.

Historically, the seakeeping of planing hulls has attracted the attention of researchers for six decades. The general knowledge regarding the dynamic of planing hulls in waves has been shaped through experimental research that was conducted in towing tanks. Some systematic studies can be found in the literature. While the body of knowledge regarding the dynamic response of the hard-chine boat is strong enough, the dynamic response of the tunneled hull form is poorly understood. We are still not sure about the method that can be used to model the dynamic responses and our understanding of the physics of the problem is immature at the present stage. This is while our knowledge regarding the calm-water performance of these vessels, as mentioned earlier, has been formed through experimental and numerical observations [12–16]. The designers in particular, need modeling of the dynamics of these boats, which can help them to fill the gaps in the design of fast tunneled boats. At the present stage, it is required to find out the effects of the aerodynamic pressure on forces, including damping and added mass ones. In addition, the very small wetted surface of the boat can increase the occurrence probability of fly-over motion.

While there is a strong experimental dataset for hydrodynamics of hard-chine hulls [17], the dataset regarding the hydrodynamic of tunneled planing hulls are scant, which forces us to employ a mathematical and numerical method for the replication of their unsteady planing motion in waves. The solution of the ideal fluid field around a structure has been widely used for modeling dynamic responses of ships/structures. In the case of planing hulls, such a methodology is mostly used for modeling steady motion. It is very complicated to implement nonlinear boundary conditions along with Kutta conditions, which govern the side edges and transom of the vessel. More importantly, aerodynamic pressure generates a significant force, which necessitates consideration of the two-phase flow around the body.

The mathematical methods are well known for their significant contribution to modeling the dynamic motions of planing hulls in waves and calm-water conditions [18,19].

However, in the case of a tunneled planing hull, the transverse section of the vessel does not have a simple shape, and as mentioned, the air flows in the tunnels. These two make the 2D + t method useless. In simple words, we need a unique theoretical simulation for a tunneled section entering the water, which is not presently available. Readers that are interested in the water entry problem can refer to [20].

The appropriate method to simulate the wave-induced motions of the high-speed tunneled hull is to solve the Navier–Stokes equations, which govern the viscous fluid field around the vessel. Turbulent two-phase flow, a mixture of air and water, is assumed to flow in a virtual water tank, and the solution of the fluid field around the vessel can be achieved. Computational fluid dynamics, CFD, provides the numerical solution of the Navier–Stokes equations. The popularity of the CFD codes in the numerical reapplication of different problems that are linked to ocean/offshore engineering can be seen in a wide range of studies that have been conducted in the last two decades [21,22]. The key strong point about CFD is that it can be used to solve the two-phase viscous flow in the fluid domain. In this condition, some physical aspects of the flow, which are neglected when an ideal flow is embarked, are considered. For example, we can model the shear stresses and turbulent development around the vessel [23], which are skipped under an ideal fluid assumption.

Going back to the last two decades, with the advances in computational mechanics and parallel processing, the application of CFD methods in simulating the planing problems has increased. The proper accuracy of CFD models in the simulation of the steady flow around planing hulls has been observed in a wide range of CFD studies (e.g., in [24,25]) which has directed boat designers to count on CFD as a promising hydrodynamic tool. Specifically, CFD models have been observed to provide a high level of accuracy in the modeling motions of the special designs of high-speed planing hulls [26–31]. CFD models have also demonstrated the ability to model the vertical motions of planing boats [32–35]. As mentioned earlier, the dynamic responses of tunneled planing hulls are not understood clearly. The numerical simulation of Navier–Stokes equations can help us to simulate the problem.

Current seakeeping studies of tunneled planing hull are conducted experimentally which has a limitation in presenting some physical details, i.e., pressure distribution, pressure and shear resistance, streamlines, etc., that are provided in the present paper. Therefore, in the present paper, the dynamic motions of tunneled planing hulls operating in head sea conditions are numerically replicated by employing state-of-the-art CFD simulations. It is aimed at providing a clear understanding regarding the dynamic motion of a tunneled planing hull that is advancing in waves at high speeds, which is lacking presently. Predicting the dynamic motions of tunneled planing hulls in waves as well as providing some insight regarding fly-over phenomenon can be applicable in designing the next generation of high-speed boats considering the vessel structure and crew safety. The rest of the present paper is structured as follows. Following the Introduction section, the equations governing the motions of the vessel and fluid motion are presented. The CFD model that was used for the numerical reproduction of the problem is described in Section 3. The model is then compared against experimental data in Section 4 of the present paper. In Section 5, the results are presented and discussed. Finally, the concluding remarks and the scope of future works are presented in Section 6.

## 2. Formulation of the Problem

Considering a three-dimensional spatial domain that is filled with water and air, a right-handed coordinate system that is denoted with Oxyz is placed in the domain. The domain represents the fluid field around a vessel that is operating in planing mode in a rough-water condition, as displayed in Figure 1. The interface between liquid and gas is the water surface. Gravity waves can propagate in the fluid domain causing the free surface to oscillate around its equilibrium condition. The water waves propagate in the longitudinal direction from one side to the other, transporting energy. They can, therefore, cause vertical motion for the vessel.

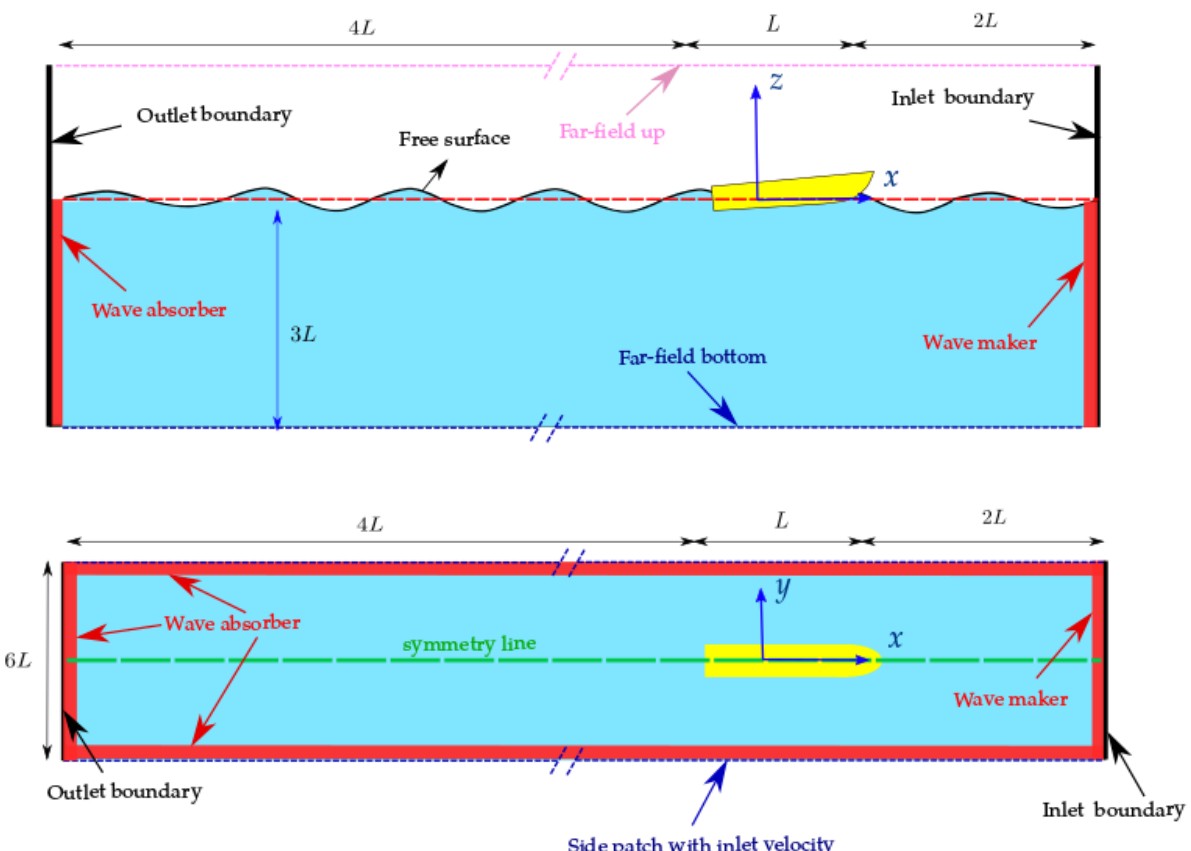

**Figure 1.** A sketch showing the problem domain. A tunneled planing craft is subjected to gravity waves that are generated by the numerical wave-maker that is located at the right end of the domain. The upper and lower panels show the longitudinal and top views of the problem. The dimensions of the domain are chosen based on the previous CFD simulations and recommendations, and their values are mentioned in Section 3 of the paper where the CFD model is described.

The fluid domain contains six patches, including two vertical, two horizontal, and two laterals. The vertical patches are the inlet and outlet boundaries. The lateral ones are the side boundaries. The bottom and upper patches are the far-field water and air boundaries. Since the fluid motion is symmetrical to the *x-z* plane passing through the centerline of the vessel, one of the side boundaries can be dropped, and the problem domain can be divided into two symmetric sub-domains. Thus, the symmetry lateral boundary can be used.

The dynamic motions of the vessel and wave motion in the fluid domain together cause changes in the linear momentum from one point to another. Therefore, the fluid problem is expected to be coupled with the dynamic motion of the vessel. See examples of the wave-induced motions that are caused by waves in [36].

The fluid motion of the air-water flow is hypothesized to be rotational, i.e., viscosity can cause vortex generation, deformations, and energy damping within the fluid domain. The flow is expected to be strongly turbulent near the walls of the vessel as the fluid velocity near its walls is relatively high. Note that the speed of the vessel, *u*, which is normalized using Froude's number, as per:

$$Fr_{\mathrm{B}} = u/\sqrt{(gB)} \tag{1}$$

is greater than 2.0. Here, *g* is the gravity acceleration constant, and *B* is the beam of the vessel.

Assuming that the fluid flow is incompressible, homogeneous, and Newtonian, the equations governing the fluid motion can be formulated. In this condition, Navier–Stokes equations, which relate to the pressure field and the velocity field, given by: $\nabla \cdot \mathbf{v} = 0$

$$\partial_t(\rho_m \mathbf{v}) + \rho_m \mathbf{v} \cdot \nabla \mathbf{v} = -\nabla p + \nabla \cdot \mathbf{Tf} \tag{2}$$

govern the fluid motion. In Equation (2), $\mathbf{v}$ is the velocity vector and $p$ is the fluid pressure. Both the velocity and pressure are unknown and need to be computed over time. $\mathbf{T}$ and $\mathbf{f}$ refer to the normal tensor and the body force vectors. The normal tensor is generated by viscosity and the body force is caused by gravity acceleration. $\mathbf{T}$ and $\mathbf{f}$ are given by

$$\mathbf{T} = (\mu_m + \mu_t)\left(\nabla \mathbf{v} + (\nabla \mathbf{v})^{\mathbf{T}}\right) \tag{3}$$

$$\mathbf{f} = \rho_m \mathbf{g}$$

In Equations (2) and (3), $\rho_m$ and $\mu_m$ are the density and dynamic viscosity of the air-water mixture at any point. $\mu_t$ is the dynamic turbulent viscosity that can be computed through turbulent flow modeling. $\mathbf{g} = [0, 0, g]$ is the gravity acceleration vector.

The density and dynamic viscosity of the mixed air-water flow are computed by:

$$\rho_m = (1 - \phi)\rho_w + \phi\rho_a \tag{4}$$

$$\mu_m = (1 - \phi)\mu_w + \phi\mu_a.$$

In Equation (4), $\phi$ is the volume fraction. Subscripts $w$ and $a$ refer to the values of air and water. The volume fraction varies between 0 to 1. This value is employed to model the two-phase, non-reacting, fluid motion. $\phi = 0$ refers to pure water, and $\phi = 1$ refers to pure air. This helps us to model the two-phase flow [37], and capture the water surface properly, especially for the case of the fluid flow around the section of high-speed boats (e.g., in [38]).

The volume fraction is transmitted within the domain. A conservation law governs the flux of $\phi$, as per:

$$\partial_t \phi + \mathbf{v} \cdot \nabla \phi = 0 \tag{5}$$

The boat is exposed to unidirectional monochromatic gravity waves that are generated at a numerical wave-maker, with an encounter angle of $\pi$. This wave-maker is located at the right side of the domain, the inlet boundary. The waves that are generated at the wave-maker lead to a head-sea condition.

As explained earlier, the wave steepness ($k_{in}A_{in}$) is set to be gentle, and thus Airy Theory governs the wave motion. No modulation instability, wave-induced turbulence, wave breaking, and wave–wave interaction is developed as the waves propagate along with the fluid domain.

The water surface elevation, $\zeta$, is given by:

$$\zeta_{in} = A_{in}\cos(k_{in}x - \omega_{in}t + \epsilon) \tag{6}$$

In Equation (6), $\omega_{in}$ and $k_{in}$ are the wave frequency and wave number, which are connected under the dispersion relation of deep-water, as:

$$\omega_{in}^2 = gk_{in} \tag{7}$$

The values of wave frequency and wavenumber are computed by:

$$\omega_{in} = 2\pi/T_{in}, \tag{8}$$

$$k_{in} = 2\pi/\lambda_{in},$$

where $T_{in}$ and $\lambda_{in}$ are the wave period and wavelength. Note that a far-field boundary condition is assumed for the bottom. Therefore, the deep-water dispersion relationship is valid.

The dynamic motion of the vessel is coupled with the first-order wave motion in the fluid domain. Two motions, heave and pitch, are assumed to be induced by the waves.

These are the vertical motions that are expected to be significant when a planing vessel is exposed to water waves.

The vertical motions of the vessel obey the rigid body law, as:

$$m\ddot{z} = \mathbf{F}.\mathbf{k} + mg \tag{9}$$

$$I\ddot{\theta} = \mathbf{M}.\mathbf{j}.$$

Here, $m$ and $I$ refer to the mass and the pitch moment of inertia of the boat. $\mathbf{F}$ and $\mathbf{M}$ are the force and moment vectors that can be computed using:

$$\mathbf{F} = \int_S (p + \sigma)\boldsymbol{n}\, dS, \tag{10}$$

$$\mathbf{M} = \int_S (p + \sigma)\boldsymbol{n} \times \mathbf{r}\, dS$$

Here, $\sigma$ is the normal stress tensor. $\boldsymbol{n}$ and $\mathbf{r}$ are the normal vector and distance vector. The force and moment contain the contribution of pressure, added mass, damping, and restoring mechanisms [39].

Note that the wave-induced motions are identified as the displacement of the centre of gravity (CG) of the vessel concerning the considered coordinate system.

## 3. CFD Model Setup

### 3.1. Numerical Technique

Fluid equations are solved over time by employing the commercial CFD code SIEMENS PLM Star-CCM+ [40]. The upstream and downstream lengths are set to be $2L$ and $4L$. The tank is filled with water up to a height of $3L$. The side patch is set to be $3L$ far from the side edge of the vessel.

The water and air are set to flow in the domain from the right end and flow out from the left end of it. The bottom and top patches of the domain are set to be the velocity inlet to replicate an open-sea condition with no bottom effect. No slip condition is prescribed for the body of the vessel. The wave absorption condition is activated at all non-horizontal boundaries except the inlet to cancel out the wave energy reflection. Also, a symmetry boundary condition is set for the symmetry surface.

The $k - \epsilon$ turbulence model with a wall function is applied. The second-order upwind scheme is used to discrete the governing equations to reduce the transaction errors All $y+$ wall treatments of wall functions are selected, the $y+$ value is set to vary between 30 to 300 near the surface of the solid body addressing the requirement of the all wall $y+$ approach [40].

As indicated in the previous paragraph, the Volume of Fluid (VOF) method is used to trace the free surface. The VOF method is combined with the High-Resolution Interface Capturing (HRIC) approach.

The time-step that is used for simulation is set to be 0.0004 s, which gives a Courant number that is lower than 0.5 during the numerical simulations. For all speeds with the same time step, the Courant number was lower than 1. This ensures the convergence of CFD simulations. The convection terms are discretized by using a second-order upwind scheme. The unsteady terms are turned into algebraic equations by employing a three-dimensional segregated implicit unsteady scheme. The Navier–Stokes equations are solved using a Pressure-Implicit with Splitting of Operators (PISO) algorithm. The dynamic motions of the vessel are computed over physical time through the dynamic fluid body interaction (DFBI) module by setting heave and pitch motions to be free. The simulations are run over 6 time periods. More than 15 cycles occur during this time period, which satisfies the recommendations of the ITTC [41].

### 3.2. Spatial Discretization

The spatial domain is discretized into small volumes, in each of which the Navier–Stokes equations are solved at every single time step. An unstructured mesh is generated as the geometry of the vessel is not simple.

To model the dynamic motions of the vessel, an inverse distance weighted mesh technique that is called morphing mesh, is used and the Morpher solver is applied in a region-wise manner. This means that each region that is associated with this motion is morphed independently. This approach is considered to be the most suitable method that can be used for modeling the fluid-solid interaction, where the solid region has a rigid displacement motion and the fluid region has a morphing motion [40].

Using the morphing approach, the body of the vessel is moved every time step under the action of fluid forces and then the motions of cells that are located near the vessel are computed.

The Surface Mesh technique is used to generate cells in the whole domain. Trimmer Mesh and Prism Mesh methods are employed to generate cells near the free surface and the rigid surface of the vessel. Finer cells are generated in the vicinity of the vessel and free surface to capture the gas-liquid interface with high resolution. Coarser cells are generated near the lower and upper patches of the domain, where the fluid motion is weakly affected by the vessel and the wave motion.

A total fo four different grids with a refinement ratio of $\sqrt{2}$. are generated. The number of cells that are generated for each case is presented in Table 1. A mesh independence study is performed to detect the most proper grid, which is presented later. A view of the generated mesh, Grid C, is shown in Figure 2.

**Table 1.** The number of cells that are generated for each case.

| Mesh | Number of Cells |
|---|---|
| Grid A | 912,343 |
| Grid B | 1,050,045 |
| Grid C | 1,291,035 |
| Grid D | 1,484,532 |

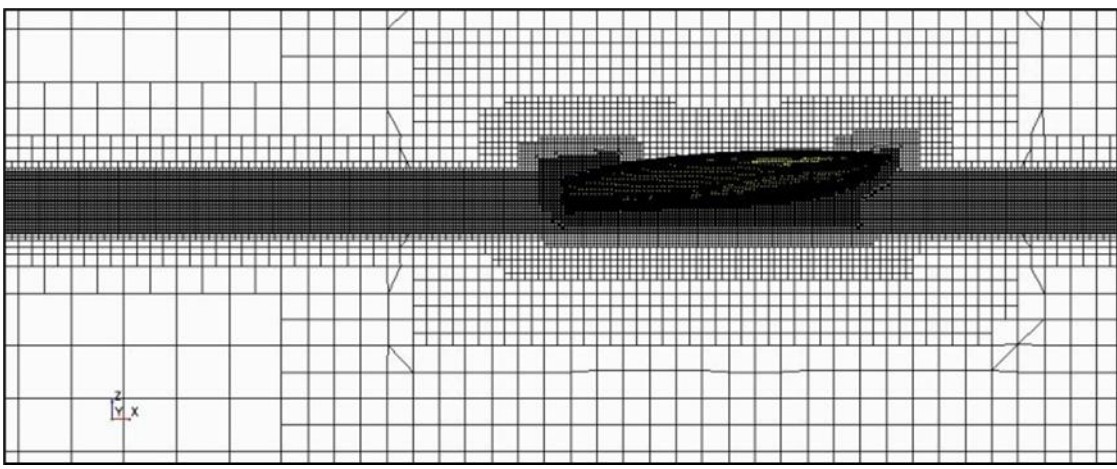

(**a**) side

**Figure 2.** *Cont.*

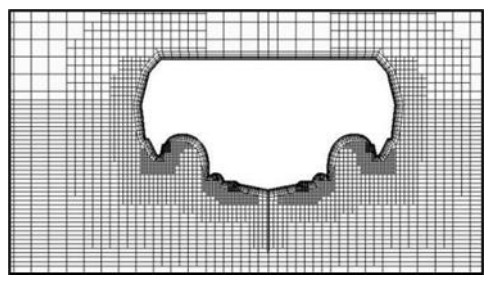
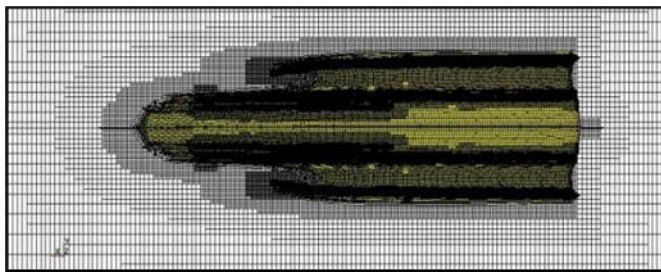

**(b)** front                                                                                     **(c)** top

**Figure 2.** Generated mesh. The upper row shows the longitudinal view of the cells around the body from a longitudinal view. The lower panels show the generated cells from side, front, and top views. The vessel that is shown here is named Model B, which is introduced in Section 3.4. The generated mesh has 1,291,035 cells (Grid C).

### 3.3. Computation of Parameters

The dynamic responses of the vessel are computed over time. The time history of the heave, pitch, and vertical acceleration at CG are sampled. The heave ($z$) and pitch ($\theta$) responses are computed by the downward zero-crossing method. The vertical acceleration ($\ddot{z}_{CG}$) is defined to be half of the difference between the crest and trough of the $\ddot{z}$ in each cycle. Note that the heave and pitch responses are normalized by dividing their values over the wave amplitude and wave steepness. Numerical data are sampled by using a frequency of 2500 Hz, which enables us to capture the data properly.

### 3.4. Planing Models and Forcing Condition

As explained earlier, two planing models are studied in the present paper. Both models have a tunneled body form and have been previously tested in towing tank experiments by researchers. The related information about these two planing designs can be found in [12,42]. The main hulls of both of the considered planing models have deadrise angles ranging between 10 to 13 degrees.

The reason for studying these two planing models is the lack of experimental data, highlighting the dynamic motions of tunneled planing hulls in waves. To the best of the authors' knowledge, no systematic experimental study has been conducted to measure the dynamic response of tunneled planing hulls in waves. There are two famous experimental studies that were carried out in the high-speed hydrodynamic tank of China Special Vehicle Research Institute, a subsidiary of the Aviation Industry Corporation of China (AVIC).

The first set of experiments was carried out by [12] in calm-water conditions. The calm-water performance of a tunneled planing hull is measured through towing tank tests. The vessel was later equipped with one, two, and three steps. It was reported that the non-stepped design had the best performance among all the cases, i.e., the resistance of the non-stepped planing trimaran hull was smaller than the other designs.

Later, another set of experiments was performed to evaluate whether air injections or a bilge keel can modify the performance of a tunneled planing vessel having large resistance [42]. The body form of this vessel is different from the one that was studied in [12]. Ma et.al. [42] also carried out a limited number of experimental tests to measure the dynamic response of the vessel in regular water waves. The experimental data they provided is the only available published research, highlighting the seakeeping of trimaran planing hulls. Importantly, the calm-water performance of the vessel that was studied in the second series of experiments is not as good as the one that was studied in the first set of experiments. The non-dimensional resistance force is larger for the case of the second set of experiments. This can be due to the different body forms, which can lead to the generation of a larger volume of water spray, and also the development of a stronger turbulent air-water mixture in the tunnels.

Resulting from our explanation, the experimental data of [42] can be used for evaluating the accuracy of the numerical model in the reproduction of the wave-induced motion of a trimaran planing hull. However, numerical simulations that are used to model the wave-induced motion of the vessel have better performance in calm-water conditions.

The wave-induced motions of the first model, Model A, are computed to evaluate the validity of the CFD model, as explained (experiments [42]). The dynamic motions of the second planing model, Model B, are simulated to understand the unsteady planing motion of the tunneled vessels (experiments performed in [12]). The calm-water performance of this model is also reported in Appendix A. The results that are presented in this Appendix A confirm the capability of the CFD model in simulations of the steady motion of Model B as well. The body plans of these models and their principal characteristics are shown in Figure 3 and Table 2. It can be seen that the dimensions of both vessels are very similar. However, their bottom shapes are different; the bottom of Model B has a convex shape.

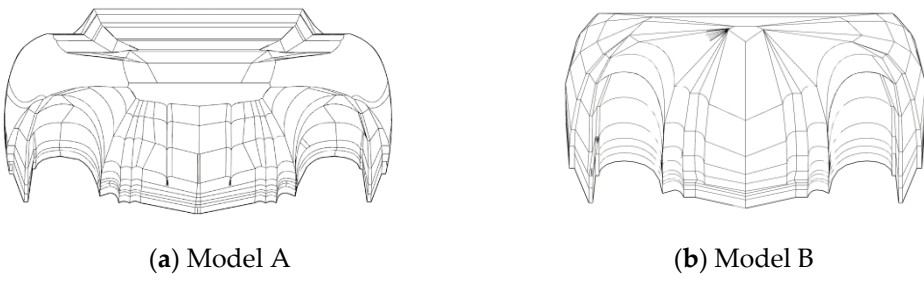

(**a**) Model A  (**b**) Model B

**Figure 3.** Body plans of the studied planing trimaran hulls [12,42].

**Table 2.** Principal characteristics of the studied models [12,42].

| Parameter | Model A [42] | Model B [12] |
|---|---|---|
| $L$ (m) | 2.4 | 2.4 |
| $L/B$ | 3.2 | 3.75 |
| $L_{CG}/L$ | 0.305 | 0.2625 |
| $V_{CG}/B$ | 0.17 | 0.156 |
| $k_{xx}/B$ | 0.36 | 0.32 |
| $k_{yy}/L$ | 0.31 | 0.27 |
| Average deadrise angle (°) | 11 | 13 |
| $m/\rho B^3$ | 0.116 | 0.149 |

### 3.5. Tests

There are two different forcing conditions that are used to induce the vertical motions of boats. The first set of conditions corresponds to the waves that are generated to trigger motions for Model A, which is used to reproduce the experiments of [42]. The data are reported in Table 3. The second forcing condition that is used in the present research is shown in Table 4. The reported waves are used to numerically generate gravity waves to induce motions to Model B.

### 3.6. Grid Independence and Uncertainty Study

A mesh independence study is performed to find the optimum number of cells of the computational domain that can be used for numerical simulation of the problem. To this end, the vertical motions of Model B are numerically simulated for $Fr_B$ of 4.7. An incoming water wave with a frequency of 0.57 Hz and a steepness of 0.03 (see Table 4) is numerically generated at the right end of the domain. The heave and pitch responses of the vessel are computed, showing that the results converge for the mesh size of 1.05 M (Grid C). A summary of the mesh independence study is shown in Figure 4, where the left

and right panels show the mesh convergence for heave and pitch responses. As seen, the coarse mesh may lead to the under-prediction of the heave and an over-prediction of the pitch responses.

**Table 3.** Forcing condition of the first set of numerical experiments.

| Wave | Wave Steepness ($k_{in}A_{in}$) (-) | $f_{in}$ (Hz) | $Fr_B$ (-) | $\lambda/L$ (-) |
|------|-------------------------------------|---------------|------------|-----------------|
| 1 | 0.10 | 1.06 | 2.1 | 0.6 |
| 2 | 0.09 | 0.95 | 2.1 | 0.7 |
| 3 | 0.07 | 0.75 | 2.1 | 0.8 |
| 4 | 0.05 | 0.72 | 2.1 | 1.2 |
| 5 | 0.04 | 0.62 | 2.1 | 1.7 |
| 6 | 0.03 | 0.55 | 2.1 | 2.0 |
| 7 | 0.02 | 0.45 | 2.1 | 3.1 |
| 8 | 0.02 | 0.44 | 2.1 | 3.2 |
| 9 | 0.02 | 0.42 | 2.1 | 3.5 |
| 10 | 0.02 | 0.41 | 2.1 | 3.7 |
| 11 | 0.01 | 0.37 | 2.1 | 4.5 |
| 12 | 0.01 | 0.34 | 2.1 | 5.4 |
| 13 | 0.01 | 0.32 | 2.1 | 6.2 |

**Table 4.** Forcing condition of the second set of numerical experiments.

| Wave | Wave Steepness ($k_{in}A_{in}$) (-) | $f_{in}$ (Hz) | $Fr_B$ (-) | $\lambda/L$ (-) |
|------|-------------------------------------|---------------|------------------|-----------------|
| 1 | 0.10 | 1.00 | 3.1-3.9-4.7-5.6 | 0.65 |
| 2 | 0.07 | 0.80 | 3.1-3.9-4.7-5.6 | 1.0 |
| 3 | 0.06 | 0.66 | 3.1-3.9-4.7-5.6 | 1.4 |
| 4 | 0.06 | 0.57 | 3.1-3.9-4.7-5.6 | 2.0 |
| 5 | 0.05 | 0.50 | 3.1-3.9-4.7-5.6 | 2.6 |
| 6 | 0.04 | 0.44 | 3.1-3.9-4.7-5.6 | 4.3 |
| 7 | 0.03 | 0.40 | 3.1-3.9-4.7-5.6 | 4.0 |

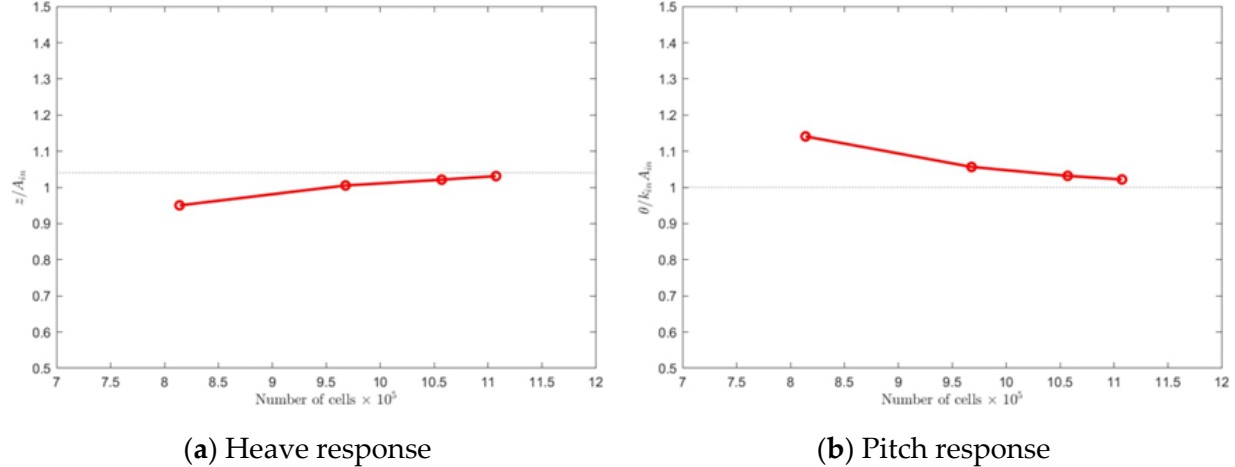

(**a**) Heave response          (**b**) Pitch response

**Figure 4.** A mesh independence study is performed to find the optimum grid size. The left and right panels show the data corresponding to the heave and pitch responses. The results correspond to the heave and pitch motions of Model B. The wavelength is 2*L*, and the Froude number is 4.7.

Also, the uncertainty analysis of the grid is presented in Table 5. Here, $R_G$ is the grid refinement ratio. Furthermore, $P_G$ is the estimated order of accuracy. GCI refers to the grid

convergence index. $U_G$ is the uncertainty of grids and $U_{SN}$ is the numerical simulation uncertainty. $|E|$ is the absolute value of the comparison error between the experimental and numerical results (of the finest grid). However, the values of $P_G$ that are less than 1 or greater than the theoretical order of accuracy ($P_{G\_th}$) determine an unreliable estimation of $\delta_{RE}$, as shown by [42]. Specifically, when $P_G < 1$, $\delta_{RE}$ is over-conservatively estimated. Instead, when $P_G > P_{G\_th}$, $\delta_{RE}$ is not reliable because $U_G$ is underestimated.

**Table 5.** Grid uncertainty analysis.

| Parameter | Grids | Grid Refinement Ratio | $R_G$ | $P_G$ | $P_{G\_LSR}$ | %$U_G$ | | %$|E|$ |
|---|---|---|---|---|---|---|---|---|
| | | | | | | GCI | LSR-GCI | |
| $\theta / k_{in} A_{in}$ | A-B-C | $\sqrt{2}$ | 1.300 | −0.756 | 0.56 | 22.907 | 21.63 | 7.07 |
| | B-C-D | $\sqrt{2}$ | 0.197 | 4.698 | | 0.256 | | 6.18 |
| $z / A_{in}$ | A-B-C | $\sqrt{2}$ | 2.400 | −2.522 | 0.829 | 12.149 | 20.12 | 7.48 |
| | B-C-D | $\sqrt{2}$ | 0.250 | 4.004 | | 0.581 | | 6.17 |

For the abovementioned reasons, Table 5 provides the results of the grid uncertainty based on the LSR (least square root) approach that was proposed by [43]. This approach is based on a least squares root version of the GCI method. Details of this method are provided in [44] and an example of application in the marine hydrodynamics field is available in [45].

### 4. Numerical Results vs. Experimental Measurements

As mentioned earlier, the objective of the present paper is to provide an understanding of the vertical motions of a tunneled planing hull operating in a head sea. There were two tunneled planing hull models, Model A and Model B, that were designed earlier and the towing tank test results have been carried out in [12,42]. Although the calm-water performance of Model B was seen to be much better compared to Model A, no experimental data highlighting the seakeeping of Model B has been published yet. Therefore, the accuracy of the CFD model in the numerical modeling of the wave-induced motion of a tunneled planing vessel is evaluated by comparing the numerical results against the experimental measurements of [42] (Model A). As mentioned earlier, the resistance of this model is higher than the other one (Model B). In addition, the calm-water performance of Model B was previously simulated by using the CFD model and compared against experimental data. The calm-water results of Model B are presented in Appendix A.

The wave-induced motions of Model A are numerically simulated over time. The forcing condition that is presented in Table 2 is used to trigger the unsteady vertical motions. Note that all the numerical tests correspond to the experiments, i.e., the experiments are numerically reproduced. All the simulations are performed at the speed of 5.7 m/s, corresponding to $Fr_B$ of 2.1.

The heave and pitch responses are computed. The results are compared against the experimental data in Figure 5. The results that were obtained using the CFD model, are seen to follow the experimental data in both panels. The numerically-predicted heave responses are in fair agreement with the experimental results of [42] at short wavelengths.

The resonance of the heave motion is seen to occur at the wavelength of ~3.5*L*, where the error of the CFD model is around ~10.15%. The heave response is seen to converge to 1.0 by the increase in the wavelength. Errors of the numerical model in computing the heave response are around ~10% in the resonance zone. The errors are around 15% at long waves. The pitch motion is seen to be resonated at the wavelength of ~4.5*L*. The CFD model has captured the peak value at a nearly shorter wavelength. Overall, the computed values of pitch response are seen to be close to the experimental data at most of the wavelengths, and they follow the experimental data. The values of errors vary between 0.08% to 5.14% at wavelengths that are shorter than 3.34*L*. For the case of longer wavelengths, where

resonance occurs, the pitch response reaches up to $3k_{in}A_{in}$, and the error of the CFD model is around ~11.54%. Overall, the results that are shown in Figure 4 prove that the present CFD model has proper accuracy in the simulation of the wave-inducted motion of a tunneled vessel that is operating in planing mode.

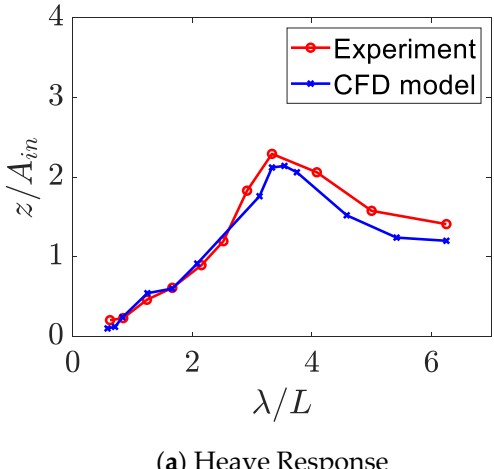

(**a**) Heave Response

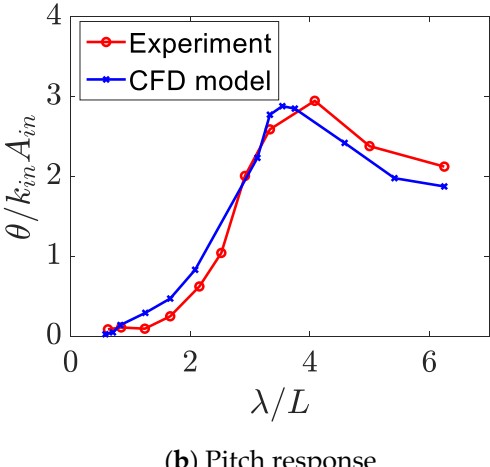

(**b**) Pitch response

**Figure 5.** Heave and pitch responses of Model A. Circle and cross markers correspond to experimental and CFD data. Experimental data are presented in [42].

## 5. Results and Discussion

The wave-induced motions of Model B are computed for different wave conditions that are reported in Table 4. The longitudinal force resisting against the vessel forward movement is also computed. The rest of the results and discussions that are presented in the present paper correspond to this set of simulations. Note that the data corresponding to the calm-water performance of Model B is presented in Appendix A. In this Appendix A, the CFD data and experimental data are compared against each other. It is shown that the CFD method can have a relatively great level of accuracy in the replication of the advancement of the vessel in calm-water. The accuracy of the CFD model in the replication of the unsteady motion of a tunneled planing hull is also evaluated in the previous section (Figure 5). Overall, the method has been seen to be accurate enough to be employed for replication of the advancement of Model B in head sea conditions. In addition, supplementary files are presented, containing recorded videos for unsteady motions of Model B, operating in water waves. Details of the videos are explained in the Supplementary Material Section.

A sample of the recorded responses of the vessel over five-wave periods, that are induced by the wavelength of $2L$ is shown in Figure 6. The results correspond to the condition the vessel advances with a Froude number of 4.7. As it can be seen, the period of the induced motion is shorter than that of the incoming wave, which agrees with the Doppler effects and the physics of the problem. The heave and pitch of the vessel are observed to have cyclic motions. The vertical motions are expected to be nonlinear. Every single cycle of heave and pitch motions is not symmetric with respect to a vertical line crossing the average point of that cycle.

The vertical acceleration is strongly nonlinear. The peak of each cycle is highly sharp. The peak value is seen to reach up to $5g$ in some cycles, which is relatively high for a vessel that is operating in the sea. Note that the wave steepness value is 0.02, which is identified as a very gentle wave condition. The minimum value of the vertical acceleration in each cycle is $-g$. This shows that the vessel experiences a $6g$ change in its vertical acceleration over a very short time. The vessel has a vertical acceleration of $-g$ over a very short time. This means that no vertical force, except the weight force, is acting on the vessel. Such a motion only occurs in the condition that the vessel comes out of the water. In this situation, the bottom surface is dry. A large acceleration occurs soon after. It occurs as the vessels re-enter the water and a large vertical impact force is generated. The center of this force is

behind the center of gravity. Thus, a large negative pitching moment emerges and pitches the vessel bow down. As a result, the pitch angle of the vessel decreases. Videos of the motions of the vessel are presented in the Supplementary Material. The file, named LF_47, includes a video of the motion of the vessel. The video demonstrates that the unsteady heave and pitch motions occur, and the vessel comes out of the water and then re-enters it. As is obvious in the video, the noticeable decrease of the pitching displacement occurs just after entering the water. In the videos, it can be seen that the rear part of the vessel enters the water first.

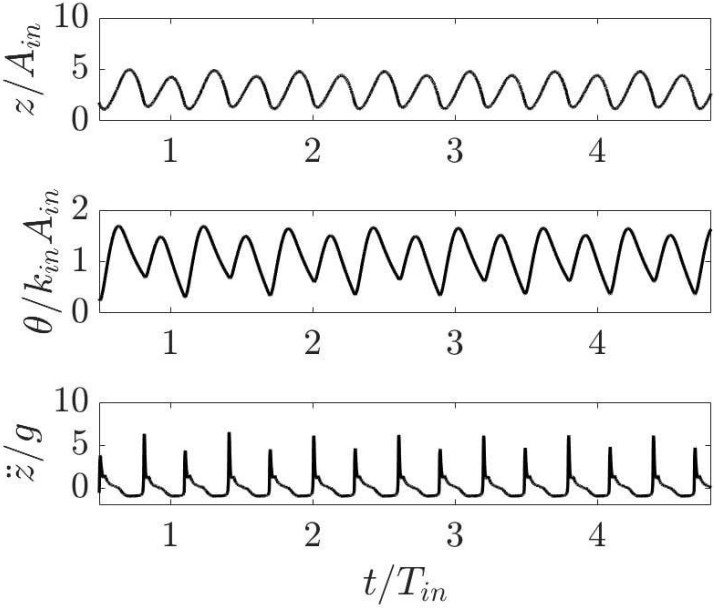

**Figure 6.** A sample of the responses of Model B to waves. The presented data correspond to 2*L* and a beam Froude number of 4.7.

Readers who are interested in an additional technical discussion on the time histories of heave and pitch responses of the hard-chine planing hulls are referred to [46].

*5.1. Responses*

The heave and pitch responses of the vessel are computed and reported in Figure 7. As seen, the resonance in the heave response occurs for all cases at wavelengths ranging between 1.4*L* to 2*L*. As it is observed, at the three lower speeds, resonance occurs at a wavelength of 1.4*L*, while it occurs at a longer wavelength at the highest Froude number (5.6). The wetted surface pattern of the vessel is expected to be the main reason for such behavior which needs to be studied in the future. The response is intensified by the increase in the speed and can reach up to 3.2$A_{in}$ in the resonance zone. For all cases, long waves induce the non-dimensional response of ~1.0., which agrees with physics, i.e., at very long waves, the vessel follows the wave motion of the water. When the incident waves are much shorter than the length of the vessel, insignificant motion is induced.

Compared to Model A, the heave response of Model B is smaller. A simulation was run for Model A for the case of a smaller beam Froude number (2.1). The resonance of heave motion was seen to occur at a wavelength of ~3.8*L*. But the heave response, corresponding to the resonance of Model A, was seen to be ~2$A_{in}$. For the case of Model B, the heave response, corresponding to beam Froude numbers of 3.1 and 3.9 is smaller than ~2$A_{in}$. It confirms that Model B performs better in water waves compared with Model A. This may be due to the different designs of these two hulls. As it was mentioned before, Model B has a convex bottom shape which can direct the water toward the transom by decreasing the pressure. This can decrease the pressure near the transom of the vessel and modifies the heave response.

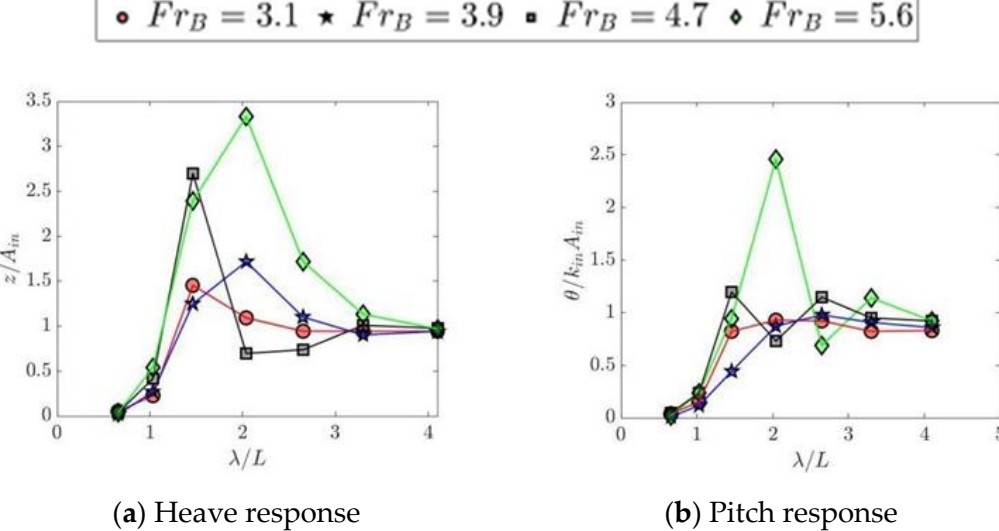

(**a**) Heave response        (**b**) Pitch response

**Figure 7.** Heave (**a**) and pitch (**b**) responses of Model B in different forcing conditions.

The pitch response of Model B, right panel, is seen to be overdamped at the small Froude numbers, i.e., no resonance occurs at the two smaller speeds, showing that the entrapped air can strongly damp the pitch motions of a vessel at a mild planing speed. At a higher speed, the wetted surface of the vessel remarkably decreases, and the volume of air supporting the vessel decreases noticeably. As a result, less pitching moment damps the angular motion of the vessel. When the speed of the vessel reaches the highest value, the pitch response emerging in the resonance zone exceeds $2k_{in}A_{in}$. The vessel has been seen following the wave slope at long waves, while short waves cannot induce any noticeable motion.

The pitch response of Model A was previously presented in Figure 5. The pitch of Model A was seen to resonate at a wavelength of ~4.5$L$. The maximum pitch response of Model A was observed to be ~3$k_{in}A_{in}$. The results of the tests that are presented in Figure 5 correspond to the Froude number of 2.1. For the case of Model B, it was seen that the pitch response is overdamped at the Froude numbers of 3.1 and 3.9, both of which are greater than 2.1. This provides evidence that the pitch responses of Model B are smaller than that of Model A. These observations demonstrate that Model B has a greater level of performance in head sea conditions. The pressure distribution pattern over the bottom surface of Model B is the likely reason for its better performance in water waves. It is interesting to note that the performance of Model A may be modified by adding one step, which can distribute the pressure over the surface by balancing its value between the front and rear body. However, it is not the aim of the present paper to investigate the effects of steps on wave-induced motion of a planing trimaran vessel. Readers who are interested in the stepped design of planing trimarans are referred to [16].

The time-averaged values of the heave and pitch motions of the vessel are computed. These values inform the mean heave/pitch displacement around which the vessel oscillates. Figure 8 shows the mean value of the heave and pitch of Model B in different wave conditions. The error bars show the amplitudes of the motion.

The mean heave of the vessel is seen to be larger than that of the calm-water condition, the dashed red line, in all cases. This means that when a tunneled planing vessel operates in waves, the nonlinear effects of water waves give rise to an extra hydrodynamic force, which pushes the vessel up. Under the action of this force, the vessel is positioned at a mean heave value, which is higher than the CG rise-up of the vessel in the calm-water conditions.

The mean values of pitch displacement are seen to be smaller than the dynamic trim angles of the vessel in the calm-water conditions. As seen, the mean pitch angle is significantly affected in the resonance zone. Interestingly, the value of pitch turns negative in some cases. For example, when the vessel operates with a Froude number of 5.6 in waves

with $\lambda = 2L$, the lower error bar reaches a negative value. In some cases, the negative value for the pitch displacement is likely to be caused by the occurrence of fly-over motion. When the planing vessel re-enters the water, a large negative pitching moment occurs, leading to a pitched-down motion. Note that after flying over the air-water interface, the stern of the vessel enters the water, causing a large negative pitching moment. An example can be observed in the Supplementary File named LF_56. In the related video, the negative pitch displacement, pitched-down motion, can be viewed after the re-entry phase.

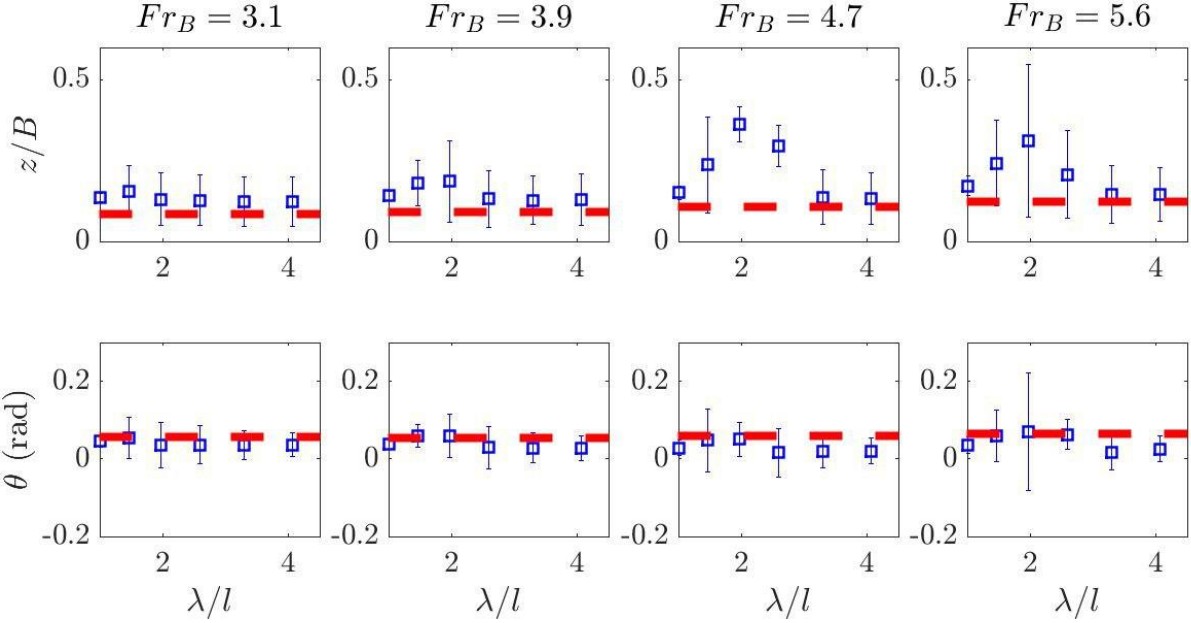

**Figure 8.** Mean values of the heave and pitch motions of Model B operating in a head sea. The dashed red line shows the calm-water performance of the model in calm-water conditions. The error bars are the amplitude of the heave and pitch motions.

In addition, it can be seen that in some other cases, the pitch angle turns negative however, the fly-over motion is not observed. For example, the pitch angle of the vessel might turn negative when its Froude number is 3.1 and the waves are 2*L* long. No fly-over motion occurs in this condition. This can be seen in the related video file (LF_31). Also, the time history of vertical acceleration of this case is presented later, showing that no fly-over motion occurs. When the vessel advances in waves with a wavelength of 2*L*, a large negative pitching moment emerges. The tunneled design of a planing hull distributes the pressure over the bottom of the vessel in a way that a large negative pitch moment might emerge when the wave crest reaches the stern of the vessel. Thus, the negative pitching moment pushes the bow of the vessel down. Note that the pressure distribution over the bottom surface of the vessel is discussed in Section 5.3.

The vertical acceleration at CG is computed for different wave conditions and is reported in Figure 9. The vertical acceleration that is caused by the long and short waves, is seen to increase by the increase in the Froude number from 2.8 to 4.3. The speed of the vessel is higher, and thus, larger vertical forces may act on its bottom surface, resulting in a larger acceleration. The amplitude of acceleration is seen to reach up to 3*g* at a Froude number of 4.7. This peak value, corresponding to a beam Froude number of 4.7, occurs when the waves are 2*L* long. In the case where the vessel operates at the highest Froude number, 5.6, the acceleration is seen to be smaller compared to the Froude number of 4.7 in some forcing conditions. The example can be seen in the case of the peak value of vertical acceleration.

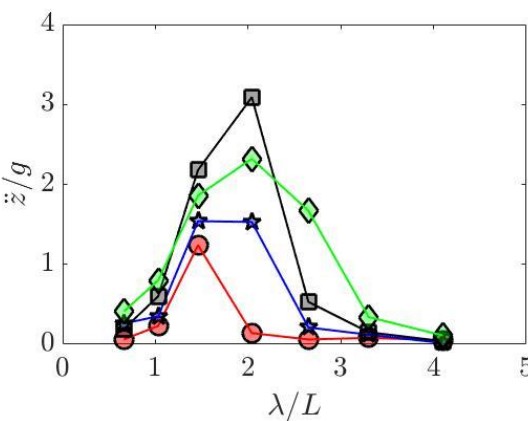

**Figure 9.** Vertical acceleration of Model B at CG.

The maximum values of acceleration are seen to occur under the excitation of waves that are 2*L* Froude number, 5.6. However, the acceleration that us related to the smaller Froude Number, 4.7, is larger when waves are 2L long. This is linked to the dynamic behavior of the vessel when fly-over motion occurs. For the case of the highest speed, the second harmonic pitch motion is relatively energetic causing two consecutive fly-over movements with different periods. This modifies the amplitude of the vertical acceleration of the boat at the highest speed. However, the vessel experienced a greater number of fly-over motions in this condition. Note that it is later shown that two consecutive fly-over motions occur at the highest speed when the incoming waves are 2*L* long.

The maximum value of acceleration emerges at longer waves as the speed is increased. At very long waves and very short waves, the vertical acceleration of the vessel converges to zero.

*5.2. Resistance in Waves*

The resistance of the vessel is sampled over time, and its amplitude is computed through a zero-crossing method. The data corresponding to resistance is displayed in Figure 10.

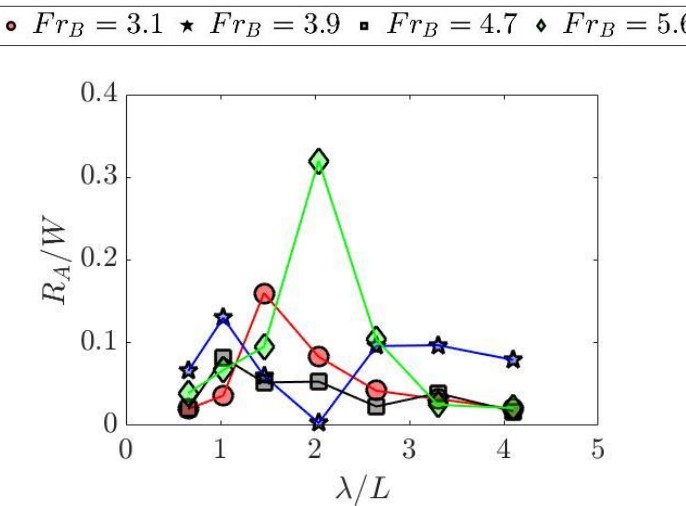

**Figure 10.** Resistance of Model B in waves.

Resistance reaches a peak value at all speeds. This value emerges at a wavelength of ~1*L* at Froude numbers of 3.9 and 4.7. The peak values of resistance at these two Froude numbers are around ~0.2W. Interestingly, the peak value of the resistance at the highest speed is ~0.38W, which is nearly two times larger than that of Froude numbers of 3.9 and 4.7. Furthermore, the peak value of the resistance corresponding to the Froude number of 5.6 emerges at a wavelength of 2*L*. The reason for such behavior needs to be investigated. In the rest of the paper, the time histories of the drag force and its components are presented to provide more understanding of such behavior.

The time history of resistance force is sampled and presented to understand the mechanisms that are causing this force. It is attempted to calculate the amount of drag that is caused by the shear stresses and pressure, separately. The time history of data, corresponding to Froude number of 4.7, and a wavelength of 2*L*, is presented in Figure 10.

As seen previously, the resistance force has a strong non-linear behavior over time. Over an exciting period, it reaches a sharp crest and then its values drop and reach zero. The zero-drag condition lasts for a period of $\sim 0.1T$. This implies that the vessel is not washed by the water, and it is above the air-water interface. Such a condition occurs when the vessel comes out of the water. This phenomenon, as defined earlier, is known as the fly-over movement and is more probable when the speed increases.

There are two different components for the resistance that are identified and presented in Figure 11 for Model B at a Froude number 4.7 in a wavelength of 2*L*. The first one is the induced drag, which is computed by integrating the hydrodynamic pressure over the wetted surface. The second one is the frictional drag which is caused by the shear stresses that is acting on the bottom surface of the vessel.

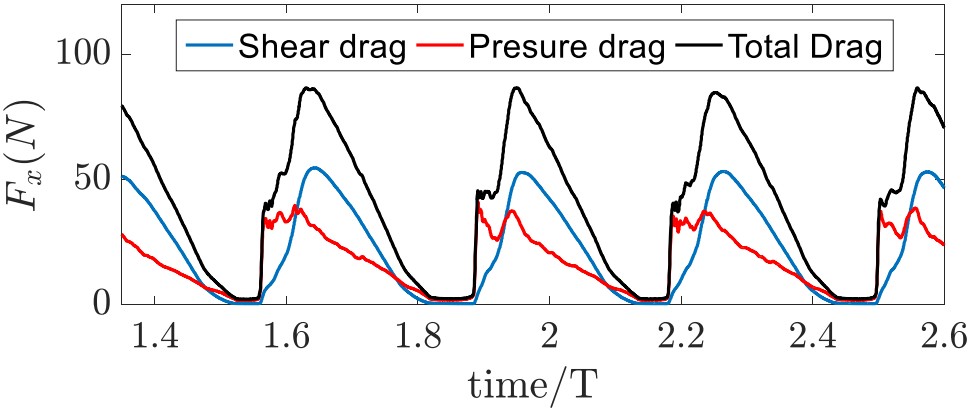

**Figure 11.** Time history of drag force, and its different components when the tunneled vessel Model B operate in waves with a beam Froude number of 4.7. The vertical motion of the body is caused by waves with a wavelength of 2L.

The drag force drastically increases and reaches a peak value. The sudden increase is observed to be linked to the sudden increase of the induced drag, the solid red curve (Figure 11). This confirms that as the vessel re-enters water and slams the free surface, a large drag force emerges. Simply stated, the solid body is impacted by the water. Thus, a larger force that is generated by hydrodynamic pressure acts on the surface in this condition. Afterwards, the contribution of the pressure drag highly decreases and converges to zero. The zero-value condition only lasts for ~0.1*T*.

Compared to the induced drag, the contribution of shear drag is seen to increase at a lower rate. However, at the instant the shear drag force reaches its peak value, the total drag force reaches its maximum value as well. When the peak drag force emerges, the value of the induced drag is notifiable. It is nearly half of the shear stress drag.

The shear drag force is expected to be dependent on the wetted area of the vessel. When the vessel reaches its lower vertical position, the shear drag is larger. Overall, the

presented results demonstrate that when the fly-over movement occurs, a very large drag force that is caused by the impact pressure emerges.

### 5.3. Fly-Over Motion

As was discussed earlier, the fly-over motion may occur when the tunneled vessel advances in a head sea condition. It was observed that the fly-over motion may lead to the generation of large induced drag forces as the vessel penetrates the water. To understand this problem more deeply, time histories of vertical acceleration are sampled and plotted. When the vessel comes out of the water, its CG has a constant vertical acceleration, which equals $-g$ as weight is the only force that is acting on it. Also, time histories of drag force are plotted to check whether the fly-over motion-induced extra drag force emerges or not.

Time histories of the resistance as well as the vertical acceleration of the vessel's left panels, and drag force, right panels, operating in waves with a length of $2L$ are displayed in Figure 12. The data that are shown in Figure 12 demonstrates that the vertical acceleration of the vessel reaches $-g$ at three higher speeds, corresponding to Froude numbers of 3.9, 4.7, and 5.6. When the vertical acceleration reaches $-g$, it does not vary over time for a very short period, e.g., $\sim 0.2T$ for a beam Froude number of 3.9. This confirms that the vessel is above the water surface, and fly-over motion has occurred, and thus only the weight force acts on it. Therefore, the vertical acceleration at CG is $-g$. Besides, the presented data for resistance confirms the occurrence of fly-over motion. It can be seen that when vertical acceleration is $-g$, the drag force is zero, i.e., the vessel is above the water and water cannot cause any friction/pressure force.

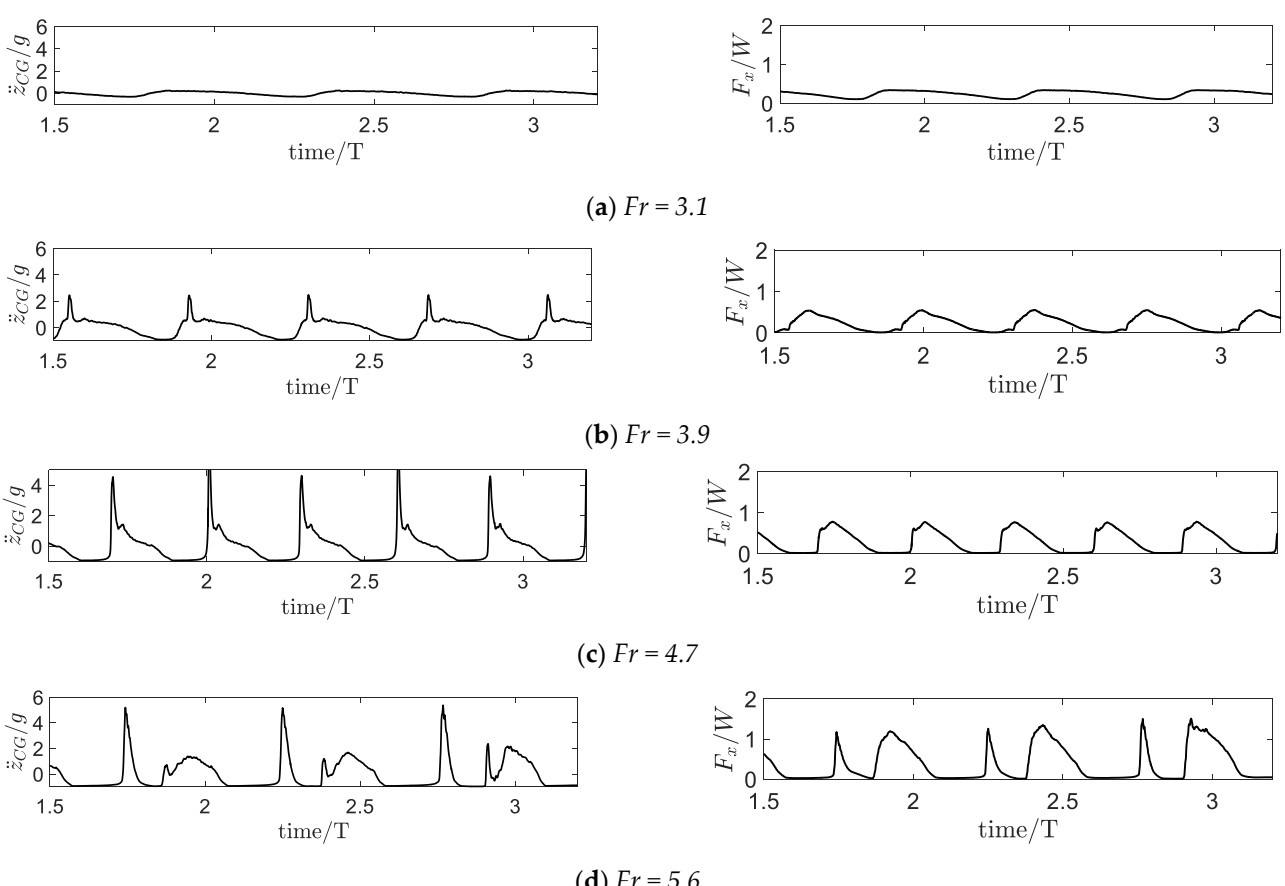

**(a)** *Fr = 3.1*

**(b)** *Fr = 3.9*

**(c)** *Fr = 4.7*

**(d)** *Fr = 5.6*

**Figure 12.** Time histories of the vertical acceleration of Model B at CG and the resistance of the tunneled planing hull in waves. (**a**–**d**) respectively denote the data corresponding to beam Froude numbers of 3.1, 3.9, 4.7, and 5.6. Note that all the motions are induced by regular waves with wavelengths of $2L$ and a wave amplitude $0.0187L$.

Interestingly, when the vessel operates at the highest speed, it has two different fly-over motions. One fly-over motion occurs just after a very large acceleration, which is around ~5$g$, and the second one occurs soon after when the vessel experiences a vertical acceleration of ~2$g$. The second fly-over motion is seen to last for a shorter time. This means that a higher frequency, which can be identified as the second harmonic, strongly involves the motion.

It seems that a higher speed can strongly affect the nonlinear behavior of the vessel that is operating in waves, and thus the acceleration energy is narrowed around two different harmonics. When the vessel operates at the highest speed ($Fr$ = 5.6), the crest to trough distance of the first fly-over motion is near ~5$g$, and the crest to trough distance of the first fly-over motion is near ~3$g$. Therefore, the average value of the CG acceleration that is found through the zero-crossing method gives a value of ~2$g$, which is smaller compared to what was found for a Froude number of 4.7. It was previously observed that the vertical acceleration corresponding to a beam Froude number of 5.6, is smaller than that of a Froude number of 4.7 when the waves are 2$L$ long. It was explained that the reason for it is the two consecutive fly-over motions that are occurring in a single exciting period.

In addition, the resistance force is affected, and thus, the pressure drag can cause two sudden increases in the time-history of the resistance. A comparison between the resistance forces of the highest speed with the other ones, confirms that the two-consecutive fly-over motions occur over a period of time.

It is very interesting to investigate the flow pattern around the vessel as it goes through the fly-over motion. It can help us to understand the physics of this phenomenon in detail. Therefore, the wetted area of the vessel, skin friction coefficient, and the pressure coefficient distribution over the bottom surface of the vessel is sampled at four different time steps. These time steps cover the different stages. The sampling is performed for four different speeds, corresponding to beam Froude numbers of 3.1, 3.9, 4.7, and 5.6. All of the results are related to the wavelength of 2$L$. Note that for the case of the lowest speed, fly-over does not occur.

Figures 13–16 show the results corresponding to beam Froude numbers of 3.1, 3.9, 4.7, and 5.6, respectively. The first row of each figure shows the position of the vessel. The second row shows the wetted area. The wetted surface pattern is identified by the distribution of volume fraction. A volume fraction of 0.0 refers to water and a volume fraction of 1.0 refers to air. A volume fraction between these two numbers includes a mixture of water and air, which mainly refers to water spray.

The third row shows the skin friction coefficient distribution over the bottom surface. This coefficient refers to the shear stresses that are generated by the effective viscosity that is acting on the wall of the vessel. It is computed through:

$$C_f = \frac{\tau_w}{0.5\,\rho_w u^2} \tag{11}$$

where $\tau_w$ is the shear stress that is caused by the air-water flow.

On the dried areas of the vessel, $C_f$ the coefficient is nearly zero, and on the washed areas, its value is non-zero. At the points where turbulence is stronger, the skin friction coefficient is expected to be larger.

The last row of the presented figures shows the hydrodynamic pressure, which is higher when the vessel impacts the free surface as it penetrates the water. This coefficient is calculated through:

$$C_p = \frac{p}{0.5\,\rho_w u^2} \tag{12}$$

As observed in Figure 13, when the vessel operates at the slowest speed, no fly-over motion occurs. This can also be seen in the Supplementary Material (LF_31). Large hydrodynamic pressure might emerge on the bow of the vessel. The skin friction coefficient is seen to be increased when the bow of the vessel comes out of the water. Its value is seen to be higher in the side bodies, where the water flow is strongly turbulent. Note that similar behavior was observed in the CFD simulations of [47]. At $t = T/2$, the crest of the wave has reached under the bow of the vessel causing noticeable hydrodynamic pressure. This results in a negative pitch motion. Then, the crest passes the transom and the wetted surface becomes smaller. The vessel is pitched down, and a negative pitch angle occurs; it is obvious in the Supplementary Material (LF_31).

Figure 14 shows the snapshots corresponding to the Froude number of 3.9 and incoming wave with $\lambda = 2L$. As was previously mentioned, the fly-over motion occurs at this speed. The occurrence of this phenomenon is obvious. The video that is presented in the Supplementary Material (LF_39) also proves that the fly-over motion occurs.

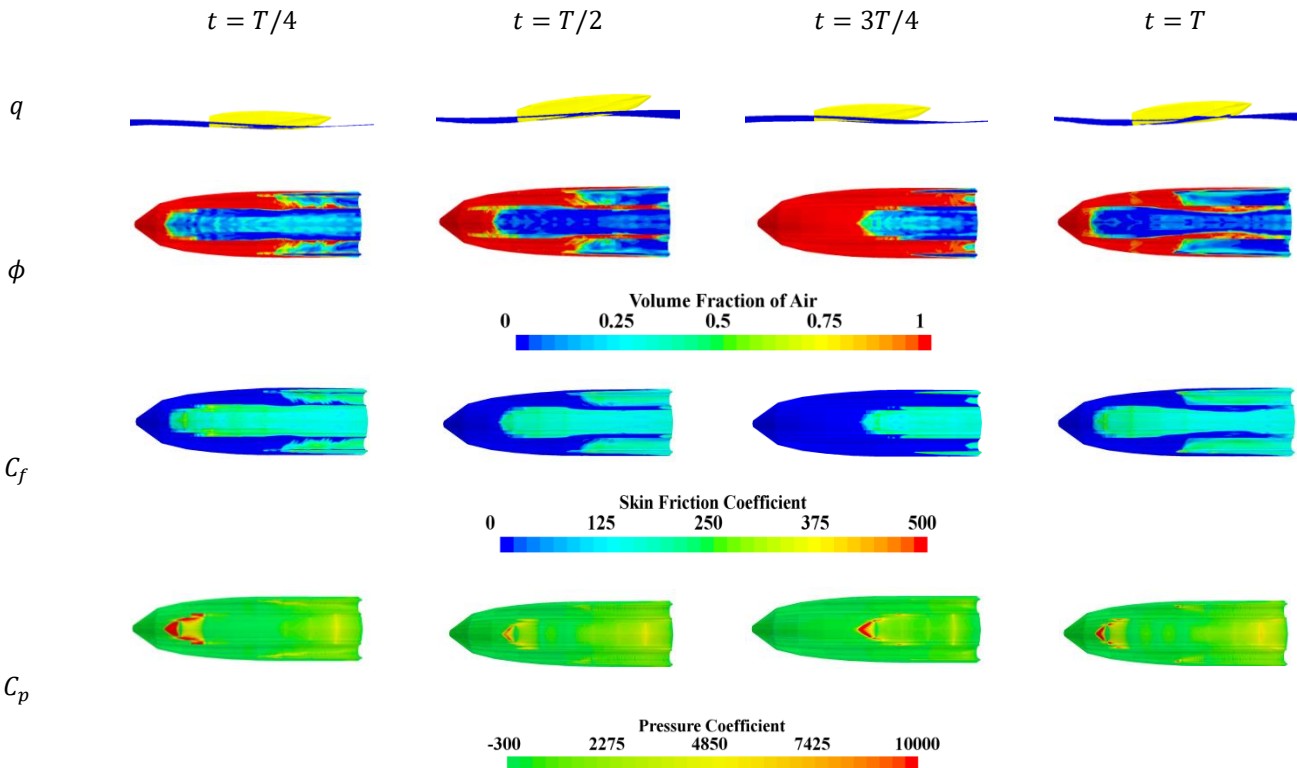

**Figure 13.** Snapshots of Model B motion during different stages. The first, second, third, and fourth rows, receptively, show the snapshots of the vessel's position to the water surface, volume fraction over, skin friction coefficient, and the hydrodynamic pressure coefficient. The results correspond to the wavelengths of 2L and a beam Froude number of 3.1.

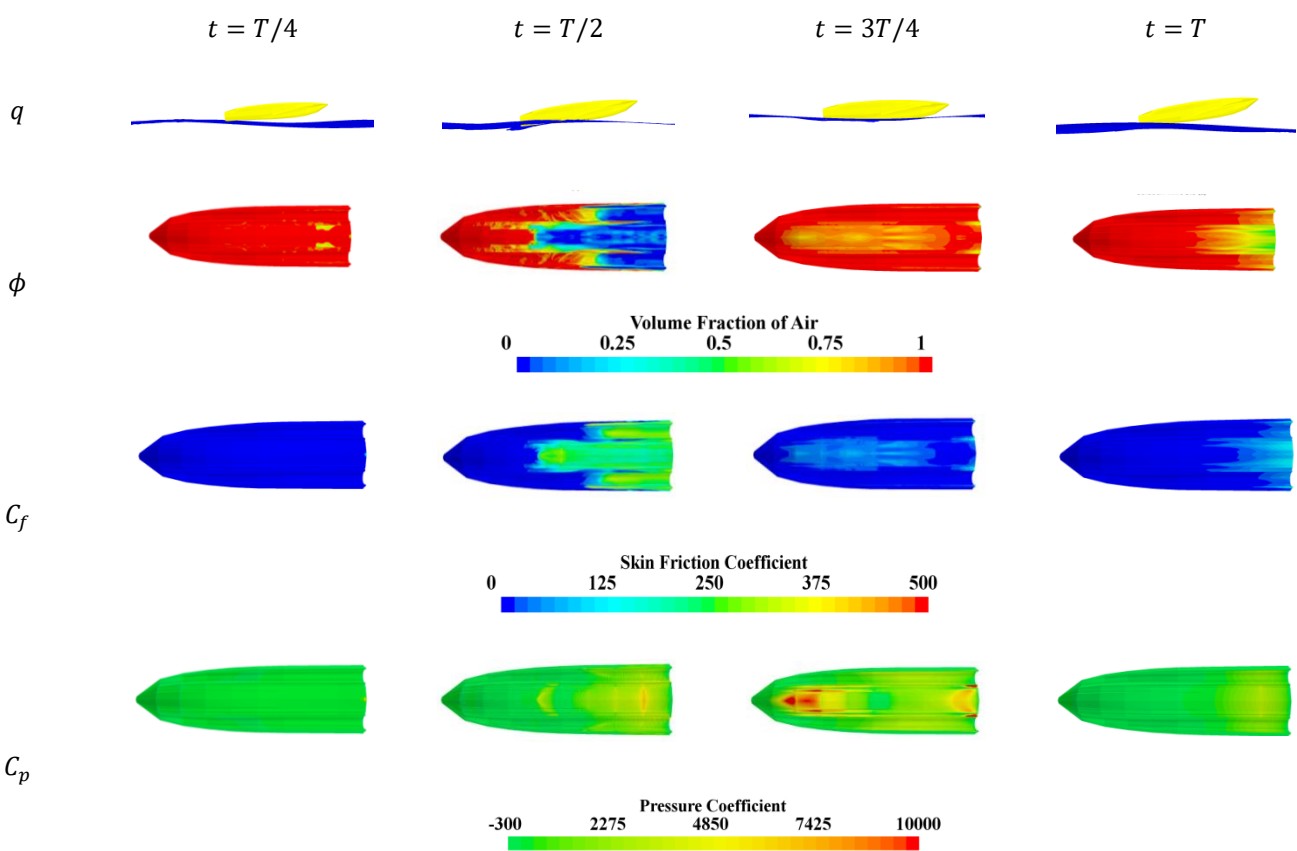

**Figure 14.** As in Figure 13, but for a beam Froude number of 3.9.

At the first snapshot, the first column, the vessel positions itself above the water surface. The surface is nearly dry and there is a low volume of the air-water mixture near the stern. The skin friction is observed to be zero on the whole body of the vessel, which agrees with the observations that were made in Figure 10. Interestingly, the hydrodynamic pressure on the whole body of the vessel is zero, which matches with the physics of the problem, i.e., no water flows under the body at this instant and thus the hydrodynamic pressure is zero over the entire bottom surface.

It can be seen that the vessel re-enters the water at the next stage, the second column. The water washes the bottom surface, mainly the stern of the vessel. A large hydrodynamic pressure emerges, and the skin friction coefficient becomes large on the side hulls, where the shear stresses are larger as turbulence kinematic energy is larger on the side bodies. The hydrodynamic pressure reaches a very large peak value at the point near the transom, which matches with previous observations (e.g., in [48,49]). Such a larger pressure is expected to be caused by the rigid body movement when the vessel falls (related discussions on water entry and the related high hydrodynamic pressure can be found in [50,51]. Note that the rear part of the vessel enters the water first.

At the next time step, the third column, the pitch angle of the vessel decreases, the heave position decreases, and the vessel skims on the water surface. In this condition, the hydrostatic pressure converges to zero. The skin friction coefficient is seen to be decreased at this stage. The hydrodynamic pressure becomes very large on the bow of the vessel. This large pressure prepares the vessel to exit the water.

At the final stage, the vessel jumps out of the water. Its bow is highly pitched up and the CG is high above the water surface. Only the rear area of the bottom is washed by water, and the skin friction is nearly zero over the bottom surface. A large pressure that was observed in the previous stage, the third column, has vanished.

The sampled data corresponding to the beam Froude number of 4.7 are displayed in Figure 15. The Supplementary file (LF_47) also shows the motion of the vessel as it is

exposed to water waves. Snapshots are presented with a shorter time step as Doppler effects lead to a shorter encounter period at this speed. The fly-over motion is observed to occur at this speed. It can also be seen in the video file that is presented in the Supplementary files (LF_47).

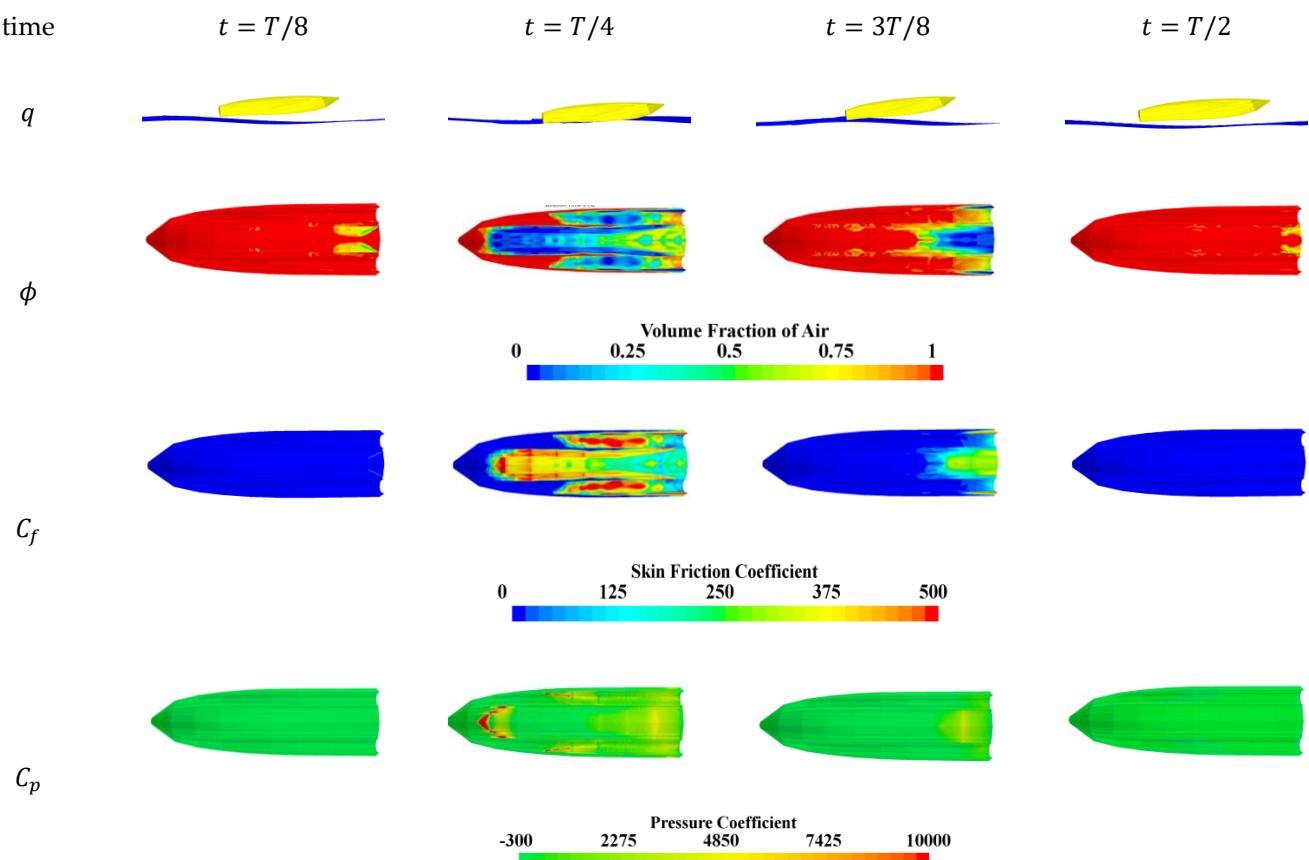

**Figure 15.** As in Figure 13, but for a beam Froude number of 4.7.

It can be seen that the vessel is above the water surface at the first and last time-steps, first and fourth columns. At the first time-step, the first column, the vessel is advancing above the water. The skin friction is zero on the surface of the vessel, signifying that no shear stress emerges on the body of the vessel as it is moving above the air-water interface. Similarly, the hydrodynamic pressure is zero on the whole body of the vessel.

As the vessel re-enters water, high pressure is seen to emerge near the bow of the vessel. This can be seen in the second column of Figure 15. The vessel is entering the water and its bow is washed by the water. Interestingly, the skin friction coefficient gets relatively large near the bow of the vessel and on the side hulls. Such a distribution of the skin friction coefficient over the bottom of the vessel is different from what was observed at two previous speeds. This implies that the gradient of the velocity near the bow of the vessel and on the side-walls is higher compared to the rear part of the vessel. The vessel enters the water with its bow, and the turbulence is strong there. In addition, water flow is prone to move toward the side hulls, which leads to the generation of vortices between the main hull and side bodies. Such a motion may result in strong skin friction on the side-bodies.

The large pressure emerging on the bow of the vessel, bounces the vessel back. Thus, its bow is highly pitched up, and the rear part of the body remains partially submerged. This can be seen in the third snapshot. The rear part of the body is wet, and the skin friction is non-zero around there.

Eventually, the vessel comes out of the water and a fly-over motion occurs; this can be observed in the fourth column of Figure 15. The vessel is located above the water surface. A small proportion of it is washed by the mixture of water-air, which is the water spray

that is trapped between the tunnels and the main hull. The third and fourth rows show that the skin friction and hydrodynamic pressure are zero on the whole body of the vessel.

Figure 16 demonstrates the snapshots that are related to the highest Froude number. The waves with $\lambda = 2L$ are generated, causing unsteady vertical motions for the vessel. Again, the snapshots are sampled with a time-step of T/8 as the speed of the vessel is relatively high. As is apparent, the fly-over movement occurs during the unsteady motion of the vessel at this speed. This can be also seen in the related Supplementary file (LF_56).

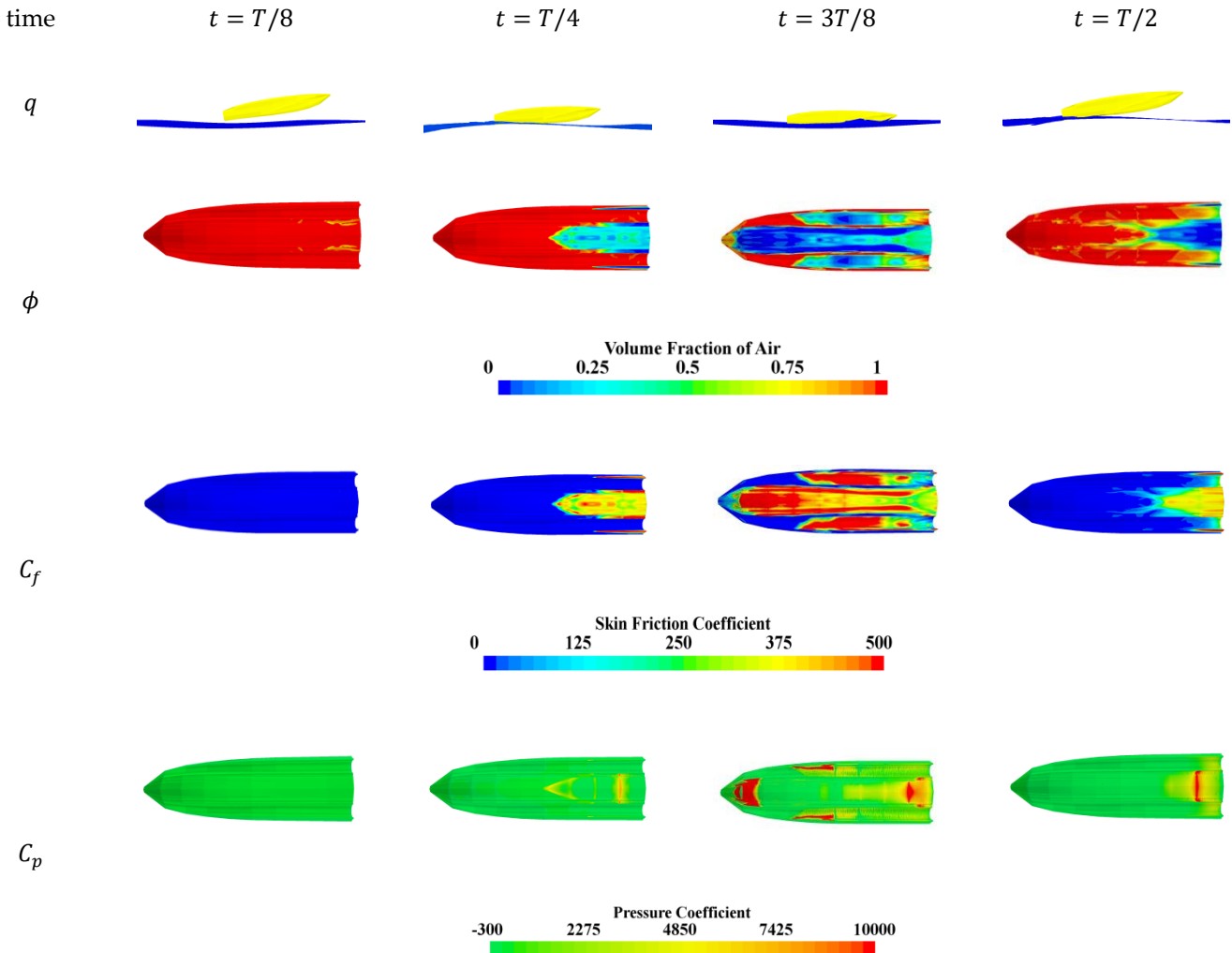

**Figure 16.** As in Figure 13, but for a beam Froude number of 5.6.

The first snapshot shows the vessel's movement above the water surface. Its bow is pitched up, and its stern is located above the water. Compared to the Froude number of 4.7, the pitch angle of the vessel during the fly-over motion is higher. The skin friction and hydrodynamic pressure are both zero at this stage.

The vessel then enters the water. This can be seen in the second column of Figure 16. The interesting point is that the stern of the vessel enters the water first. This is in contrast with what was observed for the previous Froude numbers. The hydrodynamic pressure is zero near the bow of the vessel, but it is non-zero near the stern. As a result, a large negative pitching moment is caused.

This phenomenon lasts over two time-steps as seen. Besides, the outer edges of the side bodies are only washed. The hydrodynamic pressure emerges on the washed area on the main body, but its value is not significant. Meanwhile, the skin friction coefficient on the washed area of the side bodies becomes very high.

First, its stern is partially washed, as can be seen in the second column of Figure 16. Clearly, only two small areas that are located near the stern of the vessel are partially washed by the water. The fall of the vessel into the water then continues for one more time-step, $t = 3T/8$. The bow of the vessel is pitched down, compared to the previous time-step, and then the vessel impacts the water surface. A high volume of spray water flows toward the surrounding free surface where large shear stresses emerge. At this time-step, an extremely high hydrodynamic pressure emerges on the bow of the vessel. Also, high hydrodynamic pressure appears on the front area of the side bodies. This leads to a fly-over motion (the last column of Figure 16), and the cycle occurs again.

It was previously mentioned that two fly-over motions occur when the vessel operates with a Froude number of 5.6 in waves that are 2L long. The evidence was observed in the presented time history for the vertical acceleration of the vessel (Figure 12). The vertical acceleration was seen to equal $-g$ over two different periods and the resistance was also seen to be zero over those periods. The snapshots that are shown in Figure 16, are presented with a time step of $T/4$. Such a time interval was not able to help us to capture the second fly-over motion properly. It was mentioned that the second fly-over motion corresponds to the second harmonic of the motion or a larger frequency. This means that the second fly-over motions last for a very short time, in comparison with the first one. Therefore, for the case of the highest speed, snapshots with a very small time interval are presented in Figure 17. Clear support for the second fly-over motion is seen in the presented snapshots. This fly-over motion is observed to occur between two time-steps, $t = 2.05T$ and $t = 2.24/T$. This supports the hypothesis that the second fly-over movement occurs in a short period of time, which is around $0.1T$.

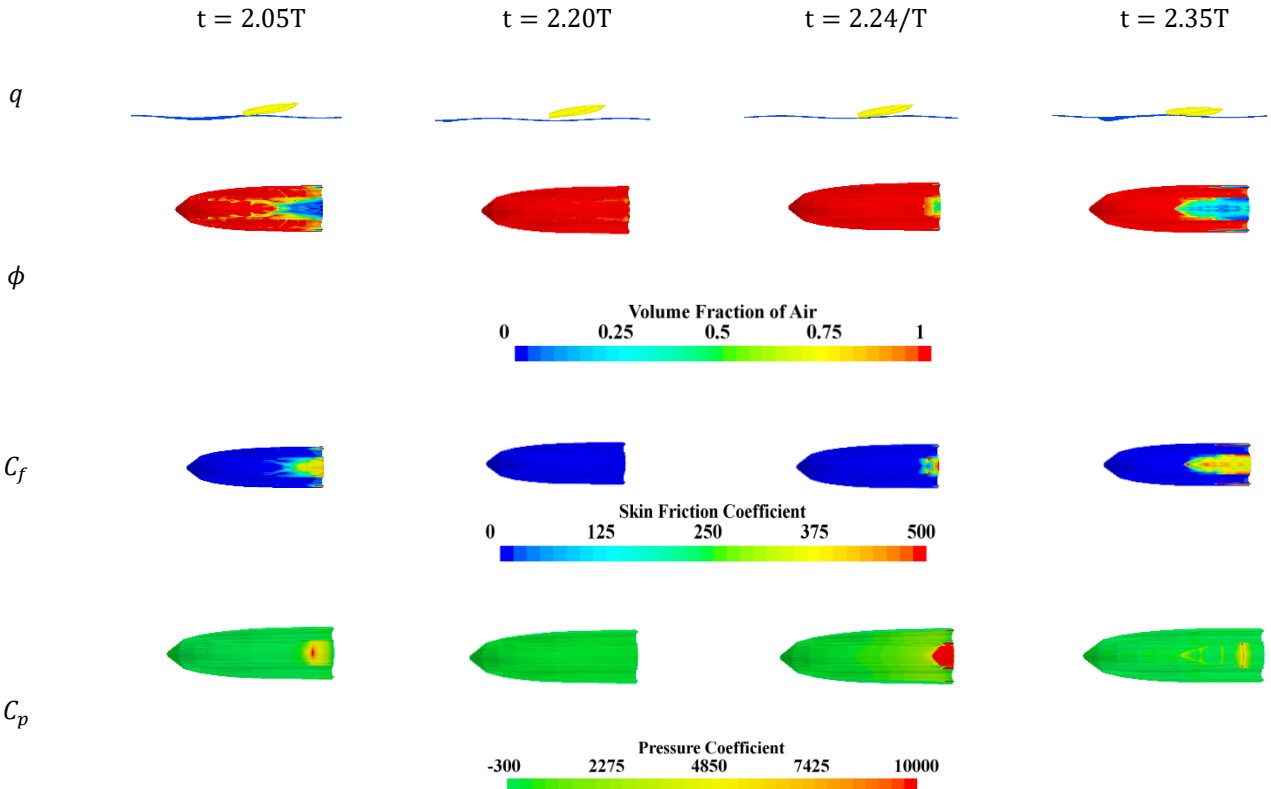

**Figure 17.** The second fly-over motion of Model B happens at the highest speed, corresponding with a Froude number of 5.6.

Interestingly, high pressure occurs near the transom before and after fly-over. As the vessel enters the water, its bow is pitched down and a large skin friction emerges near the transom. The pressure is non-zero on the side hull. When the vessel is pitched down, the pressure noticeably decreases. The skin friction gets larger on the main hull, and the

fluid motion around the vessel leads to larger skin friction on the side hulls compared to the main hull. Then, the vessel locates at a negative pitch angle, which is not shown in Figure 17. The snapshots that are presented in Figure 17 can fit between time-steps $T/8$ and $3T/8$ in Figure 16. This means that the presented snapshots demonstrate that a second fly-over motion occurs after the first fly-over motion, happening before the vessel reaches a negative pitch displacement.

To improve the understanding of fluid flow around the hull during the unsteady motion of the vessel, streamlines are also sampled and presented in Figure 18. The presented results correspond to a Froude number of 5.6 and waves with a wavelength of $2L$. The snapshots that are shown in Figure 18 cover the wave-induced motion of the vessel in a wave period. Streamlines of air and water flow are marked with red and blue colors. This helps to understand the flow behavior.

As is evident in Figure 18, when the vessel flies over the water surface, only airflow streamlines exist (Figure 18a,b). This again confirms that the vessel's bottom surface is dry during the fly-over motion. During the fly-over motion when the vessel moves downward, the streamlines deviate from the center-line (Figure 18b). This is more significant in the near transom region. The vessel is close to the water surface and moves forward at high speed. Hence, the bottom surface drives the airflow toward the edges of the vessel. This can be also viewed as a preparation stage for the vessel to re-enter water (pre-water entry stage).

After flying over the water surface, the vessel re-enters the water surface with a positive pitch angle (it was previously observed before). As the water enters, high-pressure areas emerge in the middle part and near transom regions (Figure 18c). This makes the fluid flow strongly turbulent and also causes the development of air vortex flow in the tunnels. The high pressure near the transom, as explained earlier, causes a large negative pitch motion. Hence, the vessel is pitched down at the next stage (Figure 18d). High pressure emerges near the bow of the vessel. Interestingly, the air vortex disperses at this stage. The vessel is pitched down and a large pressure occurs near the bow. Thus, air can flow under the vessel and the airflow vortex vanishes.

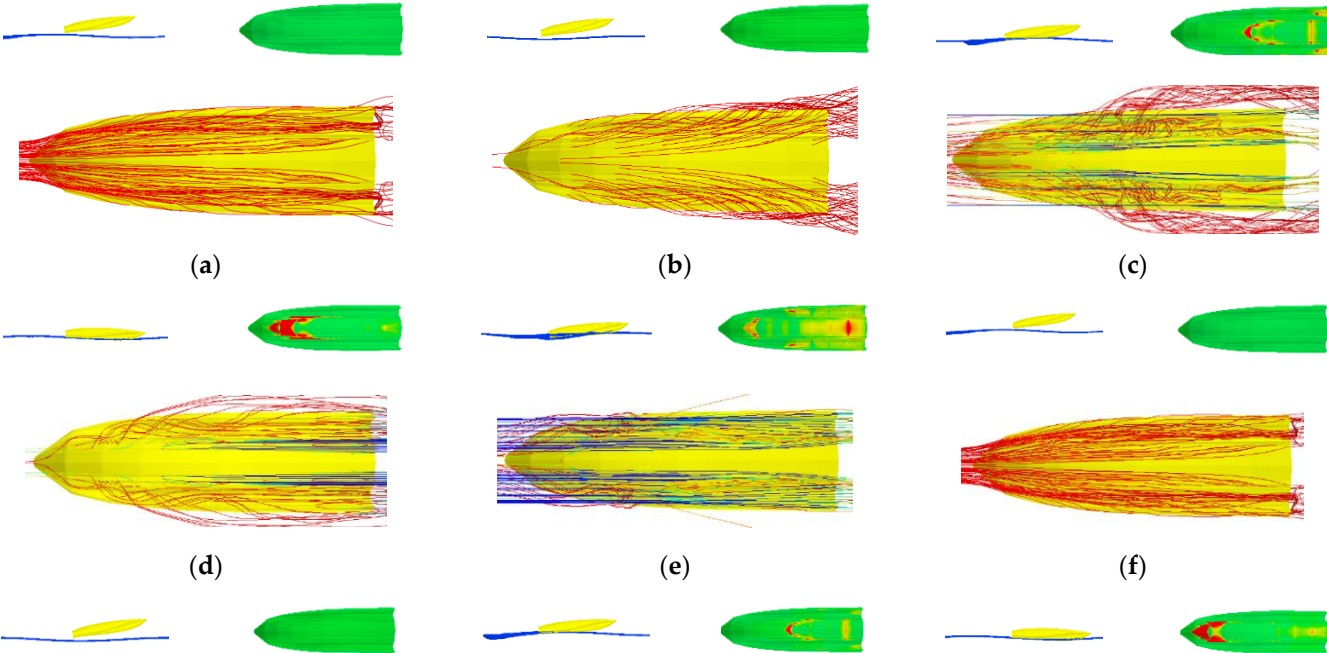

**Figure 18.** *Cont.*

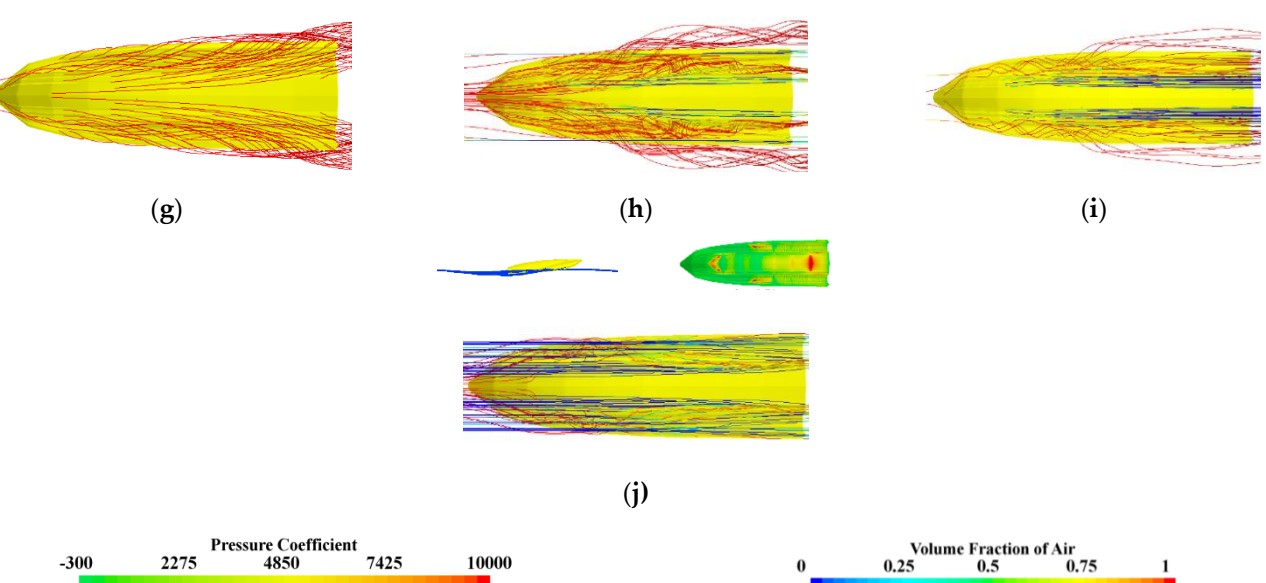

**Figure 18.** Snapshots of the air and water streamlines around the bottom surface of the Model B. The presented snapshots correspond to the motion of the vessel in wavelengths of 2*L* with a beam Froude number of 5.6. The panels respectively refer to the data that were sampled at (**a**) $t = T/10$, (**b**) $t = 2T/10$, (**c**) $t = 3T/10$, (**d**) $t = 4T/10$, (**e**) $t = 5T/10$, (**f**) $t = 6T/10$, (**g**) $t = 7T/10$, (**h**) $t = 8T/10$, (**i**) $t = 9T/10$, and (**j**) $t = T$. Each panel shows the position of the vessel with respect to the water (up on the left side), hydrodynamic pressure distribution over the bottom surface (up on the right side), and the streamlines (down).

When the vessel re-positions itself at a positive pitch angle, a large pressure emerges near the transom (Figure 18e). The vessel is moving upward with a relatively large acceleration at this stage. No air vortex emerges in this case. Note that the vessel experiences a water-exit stage, and the large pressure shifts toward the transom while the vessel moves upward. Thus, an air vortex is not generated.

Later, the vessel comes out of the water and flies over the water surface (Figure 18f,g). No water streamlines can be detected in this case. Also, no air vortex is observed. The vessel re-enters the water and again a large pressure near the transom and in the middle part of the body emerges (Figure 18h). This again leads to the generation of an air vortex in tunnels. The bow is then pitched down, and a large pressure occurs near the bow. The air vortex disperses at this stage (Figure 18i). This is exactly similar to what was observed before (in Figure 18d). The vessel locates at a positive pitch angle and is again ready to jump out of the water (Figure 18j).

Generally, we expect the airflow to damp the motion of the vessel. However, when the air vortex occurs, air cannot contribute to the damping of the motion. Instead, the vortex power might trigger a larger motion and increase the pitch motion. The air vortex is expected to occur when the vessel enters the water at a relatively large speed. At this time, a large pressure area emerges near the transom. When the vessel is pitched down, a large pressure might emerge near the bow of the vessel, which pushes the vessel up and might decrease the wave forces instantly.

All in all, the presented sampled data in this section showed that as the speed increases, the fly-over motion becomes more severe, i.e., the vessel is positioned at a higher level with a higher pitch angle as it flies over the air-water interface. The higher longitudinal speed of the vessel causes the bow of the vessel to be pitched down, and it slams into the water as its stern reaches water again. A very large hydrodynamic pressure emerges on the sides and bow of the vessel, leading to very large acceleration. Thus, a second fly-over motion occurs, during which the vessel is located at a lower level, compared to the previous fly-over movement. This movement confirms that the motion of the tunneled hulls becomes strongly non-linear and the energy of motion is divided between two dominant harmonics.

When fly-over motion occurs two times with different frequencies, the hydrodynamic pressure can lead to a very large drag force. Although a tunneled vessel is subjected to a significant value of the added resistance during the fly-over motion at high speeds, it has a superiority in comparison to the conventional hard chine planing vessels that have a lower vertical acceleration at the same speed (see [34]) and an overall reduction of the resistance, at least in calm-water at very high speed, as mentioned by [52]. The air flow between the tunnels cannot damp the motion of the vessel properly during the water entry stage. The air vortex is likely to be caused by large hydrodynamic pressure emerging near the transom. A stepped bottom design might distribute the hydrodynamic pressure more evenly. This might lead to modulating the motion of the vessel by decreasing the occurrence probability of the fly-over motion. Therefore, it is highly recommended to consider the dynamic response of the stepped hull design of tunneled planing craft in future studies.

## 6. Conclusions

Tunneled planing hulls can smoothly operate in calm-water conditions as the air flowing between the side-body and the main hull leads to a slightly larger lift coefficient. But air entrapping can influence the hull motions which becomes more important in real sea conditions. In this study, the hydrodynamic performance of the tunneled planing hulls in regular waves has been simulated using a commercial URANSE solver. The simulation results have been validated and verified by comparing against previously published experimental data.

The results showed that tunneled planing hull motions (heave and pitch) and vertical acceleration are signified by the increase in speed. Interestingly, the pitch displacement of the vessel was seen to become negative in some cases because of the fly-over motion.

During the fly-over motion, the tunneled boat was seen leaving the water and got entirely dry as it pitched up. At this stage, the vertical acceleration was found to be constant, being equal to $-g$, and drag (both hydrodynamic pressure and shear stresses) was zero. Then, as the vessel impacted water, the hydrodynamic pressure near the bow was seen to abruptly increase, inducing a very large instant drag. The shear stresses were seen to be very large in the side hulls of the vessel at this stage. This showed that the water flow becomes highly turbulent in the sidewalls, causing a large drag force over there. Moreover, when the vessel was ready to jump out of the water, the hydrodynamic pressure was seen to be very large near the transom.

For the case of the highest considered speed, two fly-over motions with different frequencies were seen. When the vessel impacts the water at this speed, the force is strong enough to direct the vessel upward again (with ~5$g$ vertical acceleration), leading to a second fly-over motion with a shorter period and lower acceleration (~2$g$). The two-consecutive fly-over motions, however, were seen to modify the vertical acceleration by distributing the energy that was related to the work that was done by water waves between two harmonics.

The results of the present paper provide an in-depth understanding of the fly-over phenomenon that can be implemented in the design of very high-speed efficient small craft where safety and crew injuries are of great importance. As a recommendation for future studies, one may simulate the wave-induced motions of the tunneled hulls in more degrees of freedom in oblique and randomized waves, where waves may stimulate noticeable coupled motions.

**Supplementary Materials:** The following supporting information can be downloaded at: https://www.mdpi.com/article/10.3390/jmse10081038/s1. Supplementary material includes six video files. Each video file shows the simulated vertical motion of Model B advancing in water waves. There are six video files that are presented. Four of these videos show the dynamic response of the vessel from a longitudinal side view. These videos are labeled with LF_NN. Here, NN refers to the beam Froude number of the vessel. All the videos correspond to the waves having a wavelength of 2*L*. Two of the videos show different views of the vessel. These two videos are labeled with 3V1 and 3V2. The first video, 3V1, shows the three-dimensional, bottom, and front views of the vessel motion. All three

views demonstrate the air-water interface and the solid body of the vessel. 3V2 shows the vertical motion of the vessel from similar views. But, for the case of the bottom view, the pressure distribution over the bottom of the vessel is shown. 3V1 and 3V2 files show the simulations corresponding to the highest Froude Number.

**Author Contributions:** F.R.: Investigation, validation, visualization, software, writing, original draft; S.T.: Formal analysis, writing—original draft, conceptualization, methodology, visualization, editing; S.M.: Investigation, writing—review and editing, A.D.: Resources, formal analysis, writing—review and editing, conceptualization, methodology, supervision. All authors have read and agreed to the published version of the manuscript.

**Funding:** The APC was funded by KTH Royal Institute of Technology.

**Conflicts of Interest:** The authors declare that they have no known competing financial interests or personal relationships that could have appeared to influence the work reported in this paper.

## Abbreviation

| | |
|---|---|
| AVIC | Aviation Industry Corporation of China |
| CF | Correction Factor |
| CFD | Computation Fluid Dynamics |
| CG | Centre of Gravity |
| DFBI | Dynamic Fluid Body Interaction |
| GCI | Grid Convergence Index |
| HRIC | High-Resolution Interface Capturing |
| ITTC | International Towing Tank Conference |
| LSR | Least Square Root |
| PISO | Pressure-Implicit with Splitting of Operators |
| URANS | Unsteady Reynolds Averaged Navier–Stokes Equations |
| VOF | Volume of Fluid |

## Nomenclature

| | |
|---|---|
| $A_{in}$ | Amplitude of the wave (m) |
| $B$ | Maximum Beam of the boat (m) |
| $C_f$ | Frictional drag coefficient (-) |
| $C_p$ | Pressure drag coefficient (-) |
| $E$ | Error of obtained results against the previous data |
| $\mathbf{F} = \left[ F_x, F_y, F_z \right]$ | Fluid force acting on the vessel in different directions (N) |
| $Fr_{\mathbf{B}} = u/(gB)^{-1}$ | Beam Froude Number (-) |
| $\mathbf{f}$ | Body force vector (N) |
| $\mathbf{f}_{in}$ | Wave frequency (Hz) |
| $g$ | Gravity acceleration (m/s$^2$) |
| $I$ | Pitch moment of inertia (Kg-m$^2$) |
| $k_{in}$ | Wave number (m$^{-1}$) |
| $k_{xx}$ | Second moment of inertia in x direction (m) |
| $k_{yy}$ | Second moment of inertia in y direction (m) |
| $L_{CG}$ | The longitudinal position of the center of gravity (CG) from transom (m) |
| $L$ | Length of bout (m) |
| $m$ | Mass of the boat (Kg) |
| $\mathbf{M} = \left[ M_\phi, M_\theta, M_\psi \right]$ | Moment vector (N-m) |
| $\boldsymbol{n}$ | Normal vector |
| $p$ | Fluid pressure (N/m$^2$) |
| $P_{G\_th}$ | Theoretical order of accuracy |
| $P_G$ | Estimated order of accuracy |
| $\mathbf{r}$ | Distance vector |
| $R_{\mathbf{A}}$ | Resistance in waves (N) |
| $R_{\mathbf{G}}$ | Converges ratio |

| | |
|---|---|
| $t$ | Time (s) |
| $T$ | Normal vector of tensor |
| $T_{in}$ | Wave period (s) |
| $u$ | Boat speed (m/s) |
| $U_G$ | Uncertainty of grids |
| $U_{SN}$ | Uncertainty of numerical simulation |
| $\mathbf{v} = \left[v_x, v_y, v_z\right]$ | Velocity vector in the fluid domain (m/s) |
| $V_{CG}$ | Vertical location of center of gravity (m) |
| $W$ | Weight of the boat (N) |
| $z, \dot{z}, \ddot{z}$ | Heave displacement (m), speed (m/s), and acceleration (m/s$^2$) |
| $\beta$ | Deadrise angle of the vessel (deg) |
| $\zeta$ | Water surface elevation (m) |
| $\theta, \dot{\theta}, \ddot{\theta}$ | Pitch angle (rad), velocity (rad/s), and acceleration (rad/s$^2$) |
| $\lambda$ | Wavelength (m) |
| $\mu_a$ | Dynamic viscosity of air (Kg/m-s) |
| $\mu_m$ | Dynamic viscosity of the Air-Water mixture at any point (Kg/m-s) |
| $\mu_t$ | Dynamic turbulent viscosity (Kg/m-s) |
| $\mu_w$ | Dynamic viscosity of water (Kg/m-s) |
| $\rho_a$ | Density of air (Kg/m$^3$) |
| $\rho_m$ | Density of the air-water mixture at any point (Kg/m$^3$) |
| $\rho_w$ | Density water (Kg/m$^3$) |
| $\sigma$ | Stress tensor |
| $\phi$ | Volume fraction (-) |
| $\tau_w$ | Shear stress caused by the air-water flow (N/m$^3$) |
| $\omega_{in}$ | Wave frequency (rad/s) |

**Appendix A. Calm-Water Performance of Model B**

The calm-water performance of Model B is presented in this Appendix A. Experimental and numerical data are both plotted and shown in Figure A1. The numerical results are obtained by using the CFD model. The water waves are set to have a height of zero. This represents a calm-water condition. The pitch angle and heave displacements of the vessel vary over time; however, they converge to steady values. The reason is that there is no exciting force, and thus the dynamic equilibrium is established in the vertical plane. This signifies that the lift force equals the weight and pitching moment that is caused by the water pressure, which is zero. Overall, the results computed through such a set-up enable us to numerically model the calm-water test of a planing vessel. The results can be compared against the experimental test. Note that these CFD runs were previously performed and presented in [53,54] For this reason, they are not presented in the main text. These results are briefly presented in this Appendix A to evaluate the accuracy of the CFD model in the computation of the calm-water operation of Model B. Discussions on the results of the calm-water condition are presented in [54].

As mentioned, the calm-water performance of Model B is presented in Figure A1. It was previously discussed that we aim to simulate the dynamic motion of this vessel in water waves since it has a lower resistance compared to another model (Model A). For the case of Model B, no seakeeping data are available, but a limited number of regular wave tests are carried out for Model A. The accuracy of the numerical model in the simulation of dynamic motion of a tunneled planing hull was previously investigated in Section 4.

Resistance of Model B is seen to be insensitive to a Froude number over the range of $1.6 < Fr_B < 2$. When increasing the Froude number, the resistance increases linearly. Then its value is not affected by the increase in the Froude number. This happens at $Fr_B > 5$. CFD data are seen to follow the experimental data, confirming that the setup is accurate enough in the computation of the resistance force acting on Model B in a calm-water operation.

Dynamic trim angle (pitch angle) is seen to decrease by the increase in the Froude number. This increase is seen to follow a linear pattern. Note that this increase is observed to occur over the range of $1.6 < Fr_B < 2.3$. Eventually, the pitch angle converges to a value between 2 and 4. Again, CFD and experimental data are seen to agree.

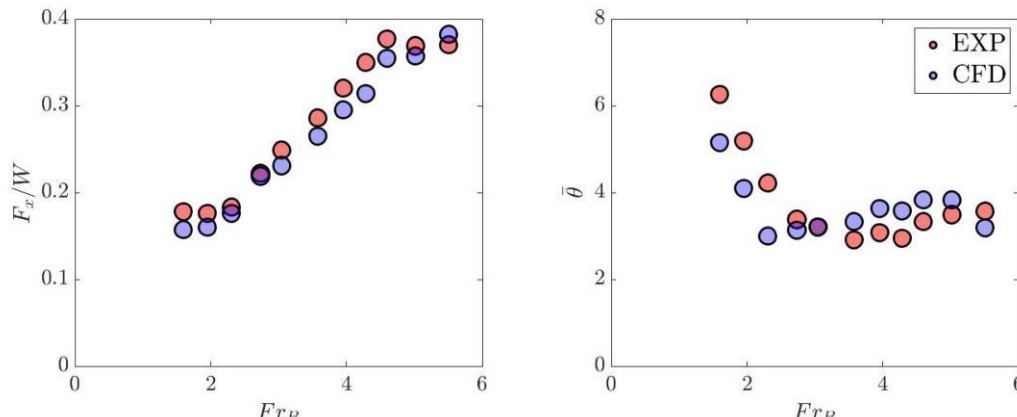

**Figure A1.** Resistance (**left**) and dynamic trim angle (**right**) of Model B in calm-water conditions. The CFD data and experimental measurements of [12] are presented in the Figure.

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
