# Peer review of "Dynamic of Tunneled Planing Hulls in Waves"

_jmse, doi:10.3390/jmse10081038_

Round 1

Reviewer 1 Report

Comments

The dynamic motions of tunneled planing hulls operating in head sea conditions are numerically replicated by employing state-of-the-art CFD simulations. It is aimed to provide an understanding regarding the dynamic motion of a tunneled planing hull, advancing in waves.

(1)  The abstract should be complete. A complete abstract should include: background, topic, method, and results. The method and result are not clear enough.

(2) There are some papers for the dynamics and response for of the tunneled planing hull. What is the difference with the existing work.

Kazemi Moghadam, H., Shafaghat, R. & Yousefi, R. Numerical investigation of the tunnel aperture on drag reduction in a high-speed tunneled planing hull. J Braz. Soc. Mech. Sci. Eng. 37, 1719–1730 (2015). https://doi.org/10.1007/s40430-015-0431-4.

Yousefi, R. , Shafaghat, R. , & Shakeri, M. . (2014). High-speed planing hull drag reduction using tunnels. Ocean Engineering, 84(JUL.1), 54-60.

Guangsheng Su, Hailong Shen, Yumin Su. Numerical Prediction of Hydrodynamic Performance of Planing Trimaran with a Wave-Piercing Bow. J. Mar. Sci. Eng. 2020, 8(11), 897; https://doi.org/10.3390/jmse8110897

(3)  It is best compared with existing work.

(4) Whether the angle of encounter needs to be considered.

(5) How the waves are simulated and what order they are.

(6) In the conclusion section, the conclusion should be concise and clear.

(7) What is the impact of the paper in terms of maritime design and operations. The analysis about it must be included in the last part of the introduction and the conclusions and if possible the abstract. It is not clear how the results presented can be used.

Reviewer 2 Report

It is my opinion that this study is advanced, comprehensive, new, and important for designers of fast boats.  I have few minor comments:

(1) Line 230: “Fluid equations are solved over time by employing a CFD code [39].”

I think [39] is not the proper reference, probably [40].  I suggest indicating here (besides the reference) the name and developer of the specific software that is applied for this study.

(2) I did not find the definition of wave-steepness, is it wavenumber times wave-amplitude?

(3) I suggest indicating the wave amplitudes in the caption of Figure 12.  I guess the amplitude is determined by the wave steepness and wavelength in table 4; however, it is important to see it explicitly with the vertical accelerations.  If my interpretation is correct, Figure 12 corresponds to case 4 in table 4, where ka=0.06 and a/L=0.01917.  Scaling the model by 5 to a L=12m boat of beam B=3.2m, the wave height is 0.46m.  At FrB=5.6 the speed is 61 knots.  To my experience vertical accelerations of 5g, while sailing 61knots at 0.5m short (24m) waves are not very large as concluded.  This means that the trimaran shape improve the seakeeping relative to a typical deep V boat.

Reviewer 3 Report

1) The Abstract is too general and mainly descriptive. In the Abstract the Authors should add some of the most important results obtained in this research (its exact values), which will highlight the novelty of the presented paper already in the Abstract. Therefore, the Abstract requires re-arrangement and addition of the most important obtained results.

2) All abbreviations, symbols and markings used throughout the paper should be presented and explained in the List of Abbreviations at the end of the paper. At the moment, some of them are missing in the mentioned List, so please correct the List of Abbreviations and add in it all abbreviations, symbols and markings used throughout the paper text.

3) Figure 3 – the figure and the figure title should stand on the same page.

4) Figure 5 – the presented validation results seem to be too short. In my opinion, the Authors should present more validation results (I will leave to the Authors which exact validation parameters should be presented). Only two figures did not look like proper validation, at least to me. Also, by observing currently presented validation results, arises a question – are the obtained differences acceptable? Which are acceptable accuracy and precision ranges (between simulation and experiment) for the similar problems from the literature? At least a discussion related to the accuracy and precision of the simulations should be added in the paper.

5) Section 3 – in the paper exists two Subsections 3.4 – please, perform correct Subsections numeration.

6) In general, English is good and understandable. However, in some sentences, paper parts and in the titles of the figures occur some obvious mistakes which should be corrected. For example, in Figure 5 title is written: “is are presented”, which is surely not correct. Another example: Line 706 – it is not “Martial” – it is Material. Please, perform careful check and corrections related to the English throughout the paper text.

7) Figure 10 – the entire figure should be placed in the same page. At the moment, the top of the figure remains on the previous page. The same is valid for Figure 17.

8) Call on the Figure 11 is missing in the paper.

9) Figures from 11 to 18 are for the Model B? It is not clear, so it should be highlighted in the paper text or in the title of each Figure.

10) Figure 13 title is presented as a part of the paper text. It should be corrected and clearly presented (as in the other figures).

11) Line 777 - supplementary file is LF_56, not LB_56. Please, correct all such obvious mistakes throughout the paper text, because they can be really annoying.

12) Lines from 862 to 868 – remove bold in the text.

13) Conclusions – in my opinion, the Conclusions section is too long, and it should be shortened. There is no need for long discussions in the Conclusions section. Also, as in the Abstract, the Conclusions section should be improved with the most important obtained results (its exact values) because it seems to be too descriptive and general, without any details obtained in the presented analysis.

Final remarks: This is an interesting paper, but it should be improved and presented more clearly before potential publication. Also, additional material was very helpful for a complete and proper understanding of many details and elements.

Round 2

Reviewer 1 Report

None.

Reviewer 3 Report

The Authors have performed all corrections/additions/improvements mentioned in my review. In relation to the validation-I completely agree with the explanations presented in the Authors answers to my comments. Now, after proper revision, I have no more concerns in relation to this paper. The paper should be published in a presented (revised) form. My congratulations to the Authors.